**Soil carbon release responses to long-term versus short-term climatic**
**warming in an arid ecosystem**
Hongying Yu[1,2], Zhenzhu Xu[1, *], Guangsheng Zhou [1, 3, *], and Yaohui Shi[1, 3]
[1]State Key Laboratory of Vegetation and Environmental Change, Institute of Botany,
Chinese Academy of Sciences, Beijing 100093, China
[2]University of Chinese Academy of Sciences, Beijing, 100049, China
[3]Chinese Academy of Meteorological Sciences, China Meteorological Administration,
Beijing 100081, China
*Authors for correspondence
**Abstract.** Climate change severely impacts grassland carbon cycling by altering
rates of litter decomposition and soil respiration ($R_s$), especially in arid areas
steppes. However, little is known about the $R_s$ responses to different warming
magnitudes and watering pulses *in situ* in desert steppes. To examine their effects
on $R_s$, we conducted long-term moderate warming (four-year, $\sim 3°C$), and short-
term acute warming (one-year, $\sim 4°C$), and watering field experiments in a desert
grassland of Northern China. While experimental warming significantly reduced
average $R_s$ by 32.5% and 40.8% under long-term moderate and short-term acute
warming regimes, respectively, watering pulses (fully irrigated the soil to field
capacity) stimulated it substantially. This indicates that climatic warming
constrains soil carbon release, which is controlled mainly by decreased soil
moisture, consequently influencing soil carbon dynamics. Warming did not change
the exponential relationship between $R_s$ and soil temperature, whereas the
relationship between $R_s$ and soil moisture was better fitted to a sigmoid function.
The belowground biomass, soil nutrition, and microbial biomass were not
significantly affected by either long-term or short-term warming regimes,
respectively. The results of this study highlight the great dependence of soil carbon
emission on warming regimes of different durations and the important role of
precipitation pulses during the growing season in assessing the terrestrial
ecosystem carbon balance and cycle.
**Key words:** Long-term warming; Precipitation pulse; Soil carbon release;
Response sensitivity; Belowground characteristics; Desert grassland.

# 1 Introduction
The global carbon (C) cycle is a critical component of the earth's biogeochemical
processes and plays a major role in global warming, which is mainly exacerbated
by the elevated carbon dioxide ($CO_2$) concentration in the atmosphere (e.g.,
Falkowski et al., 2000; Carey et al., 2016; Ballantyne et al. 2017; Meyer et al.,
2018). Soil respiration ($R_s$), mainly including the respiration of live roots and

microorganisms, is a key component of the ecosystem C cycle as it releases $c$. 80 Pg of C from the pedosphere to the atmosphere annually (Boone et al., 1998; Karhu et al., 2014; Liu et al., 2016; Ma et al., 2014; Schlesinger, 1977). The effects of both soil moisture and temperature on $R_s$ processes and the eco-physiological mechanism are reported extensively; however, it is not well known how soil moisture modulates the response of $R_s$ to changes in the duration and intensity of warming, particularly in arid and semiarid areas, where water and nutrients are both severely limited (e.g., Dacal et al., 2019; Fa et al., 2018; Reynolds et al., 2015; Ru et al., 2018).

The desert steppe of China is $c$. 88 million $hm^2$, accounting for 22.6% of all grasslands in China, and is located in both arid and semiarid areas. More than 50% of the total area of the steppe is facing severe degradation in terms of the decline of community productivity and soil nutrient depletion, primarily due to improper land use, such as over-grazing and adverse climatic changes, including heat waves and drought stresses (Bao et al., 2010; Kang et al., 2007). Global surface temperature—mainly caused by the anthropogenic $CO_2$ increase—is expected to increase from 2.6 to 4.8°C by the end of this century, accelerating land degradation (IPCC 2014; 2019). Moreover, the desert steppe ecosystem with low vegetation productivity is vulnerable to its harsh environmental conditions, such as scarce precipitation and barren soil nutrition. For instance, water deficit and heat waves during the growing season can markedly decrease plant cover and productivity in this arid ecosystem (Hou et al., 2013; Luo et al., 2018; Maestre et al., 2012; Yu et al., 2018).

Numerous studies have shown that soil temperature and moisture are the two crucial factors that mainly control $R_s$; however, it is not well known how soil moisture status mediates the response of $R_s$ to the changes in the duration and intensity of climatic warming. Soil temperature is the primary factor driving temporal $R_s$ variations (e.g., Carey et al., 2016; Gaumont-Guay et al., 2006; Li et al., 2008; Wan et al. 2005). Generally, $R_s$ is significantly and positively correlated with soil temperature when soil moisture is ample (Curiel et al., 2003; Jia et al., 2006; Lin et al., 2011; Reynolds et al., 2015; Yan et al., 2013). In general, the seasonal variations of $R_s$ coincide with the seasonal patterns of soil temperature (Keith et al., 1997; Lin et al., 2011; Wan et al., 2007). For instance, Lin et al. (2011) reported that 63 to 83% of seasonal variations of $R_s$ are dominantly controlled by soil temperature. Diurnal $R_s$ variations are highly associated with variations in soil temperature (Drewitt et al., 2002; Jia et al., 2006; Song et al., 2015). Soil respiration, according to previous studies, is expected to increase with soil water content (SWC) (e.g., Chen et al., 2008; Song et al., 2015; Wan et al., 2007; Yan et al., 2013). However, when the SWC exceeds the optimal point to reach saturated levels, $R_s$ decreases (Huxman et al., 2004; Kwon et al., 2019; Moyano et al., 2012; Moyano et al., 2013; Wang et al., 2014; Yan et al., 2018). In a study conducted in

a tall grass prairie, water addition dramatically increased soil $CO_2$ efflux (Liu et al.,
2002). Liu et al. (2009) showed a significant $R_s$ increase after a precipitation pulse
in a typical temperate steppe. Therefore, in arid and semiarid regions, where soil
water is limited, the SWC may control $R_s$, and regulate the warming effect (Chen
et al., 2008; Curiel et al., 2003; Shen et al., 2015). Furthermore, the effect of
watering pulses depends on the pulse size, antecedent soil moisture conditions, soil
texture and plant cover (Cable et al., 2008; Chen et al., 2008; Shen et al., 2015;
Hoover et al., 2016). For instance, the results by Huxman et al. (2004) showed that
different precipitation pulses have different effects on carbon fluxes in these arid
and semiarid regions; and Sponseller (2007) indicated that $CO_2$ efflux increases
with storm size in a Sonoran Desert ecosystem.
A previous study has indicated that the short-term (two-year) warming (2°C)
did not affect significantly respiration rate during the growing season (Liu et al.,
2016). However, there is limited information about the long-term (four-year)
warming effects on $R_s$ and the underlying mechanism. In this current study, we
expect that the long-term (four-year) warming may have more profound effects $R_s$
relative to previous two-year short term; and the underlying mechanism under
longer term warming condition, and the role of soil water status to $R_s$ responses to
climatic warming are also required to be explored further. Thus, in the present study,
we used a randomized block design with three treatments: control (no warming, no
watering), long-term moderate warming (four years extending from 2011 to 2014),
and short-term acute warming (one year in 2014). Moreover, a watering pulse
treatment (a full irrigation to reach field capacity) was also established. We present
the following hypotheses: (i) both long- and short-term climatic warming can
reduce soil $CO_2$ efflux, in which decreased soil moisture plays a key role in
reducing $R_s$ in the arid ecosystem; and (ii) the dynamics of $R_s$ in the water-limited
ecosystem can be driven mainly by the combination of soil temperature and soil
moisture, and soil moisture can modulate the response of $R_s$ to warming.

## 2 Methods and materials

### 2.1 Experimental site

The experiment was conducted in a desert steppe about 13.5 km from Bailingmiao
in Damao County (110°19′53.3″E, 41°38′38.3″N; 1409 m above sea level),
situated in Nei Mongol, Northern China. This area is characterized by a typical
continental climate. The mean annual temperature of this area was 4.3°C with a
minimum of –39.4°C and a maximum of 38.1°C from 1955 to 2014. The mean
annual precipitation is 256.4 mm and approximately 70% of the annual
precipitation is distributed in the growth season period occurring from June to
August (Supplementary Figure S1). According to Chinese classification, the soil
type is called "chestnut" (Calcic Kastanozems in the FAO soil classification) with
a bulk density of 1.23 $g \cdot cm^{-3}$ and a pH of 7.4. The area has not been grazed since
1980; the dominant species is *Stipa tianschanica* var. klemenzii, accompanied by
*Cleistogenes squarrosa*, *Neopallasia pectinata*, *Erodium stephanianum* and
*Artemisia capillaris* (e.g., Hou et al., 2013; Ma et al., 2018).

**2.2 Experimental design**
The warming experiment used a randomized block design. The long-term moderate
warming plots were exposed to long-term warming from early June to late August
(the growing season) for four years (2011–2014), while short-term acute warming
was manipulated only during the growing season (June to August) in 2014.The
targeted increases in temperatures relative to ambient temperature (control) are
around 3°C and 4°C under the long-term moderate warming (four-year), and short-
term acute warming regimes (one-year), respectively. Watering pulse treatments
were conducted in August in 2014 and 2017. The control plots received no
additional treatments of either temperature or water (they were recognized as
warming or watering control treatments). All of the warmed plots were heated 24
h/day by infrared (IR) lamps (1.0 m long) (GHT220-800; Sanyuan Huahui Electric
Light Source Co. Ltd., Beijing, China) at 800 W during growing seasons in the
experimental years (2011–2014). The IR lamp heights above the ground were 1.5
m and 1.0 m in moderately and acutely warmed plots, respectively. This facility
can effectively mimic different climatic warming regimes in field *in situ*, as
previously reported (e.g., Hou et al., 2013; Ma et al., 2018; Yu et al., 2018). The
watering pulse plots were fully irrigated to field capacity to simulate a watering
pulse on August 19, 2014, and August 14, 2017. For the field warming facility, to
simulate the shading effects, the control plots were installed with a "dummy"
heater similar to those used for the warmed plots. There were a total of 15
experimental plots (2 m × 2 m) arranged in a 3 × 5 matrix with each treatment
randomly replicated once in each block across three experimental blocks; a 1 m
buffer for each adjacent plot was made.

**2.3 Soil temperature and moisture**
At the center of each plot, a thermocouple (HOBO S-TMB-M006; Onset Computer
Corporation, Bourne, MA, USA) was installed at a depth of 5 cm to measure the
soil temperature, and a humidity transducer (HOBO S-SMA-M005; Onset
Computer Corporation, Bourne, MA, USA) was installed at a depth of 0 to 20 cm
to monitor the soil moisture (v/v). Continuous half-hour measurements were
recorded by an automatic data logger (HOBO H21-002; Onset Computer
Corporation, Bourne, MA, USA).

**2.4 Soil respiration**
The soil respiration was measured with a Li-8100 soil $CO_2$ Flux System (LI-COR
Inc., Lincoln, NE, USA) with the $R_s$ chamber mounted on polyvinyl chloride (PVC)

collars. Fifteen PVC collars (10 cm inside diameter, 5 cm in height) were inserted into the soil 2 to 3 cm below the surface. They were randomly placed into the soil in each plot after clipping all plants growing in the collar placement areas. The collars were initially placed a day before measurements were begun to minimize the influence of soil surface disturbance and root injury on $R_s$ (Bao et al., 2010; Wan et al., 2005). Respirations for the control and all of the warmed plots were measured from 6:00 a.m. to 6:00 p.m. on July 7 and 8 and August 18, 19, 20 and 21, 2014. The $R_s$ for watering pulse treatment was measured after the water additions on August 19, 2014, and August 14, 15, 16 and 17, 2017. To stabilize the measurement, $R_s$ was measured only on the selected typical days (i.e., mildly windy, sunny days). The $R_s$ in all plots was measured once every 2 h on that day and each measurement cycle was finished within 30 min to minimize the effects of environmental variables, such as temperature and light. Thus, a total of six measurement cycles was completed each day. The soil water content (SWC, (0–20 cm soil depth) in watering plots was measured using the Field Scout TDR 300 Soil Moisture Meter (Spectrum Technologies, Inc., Aurora, IL, USA).

**2.5 Belowground biomass and related soil characteristics**

Soil samples of 0 to 10 cm in depth were taken from each collar after the $R_s$ measurements and then passed through a 1 mm sieve to separate the roots. The roots were washed and oven-dried at 70°C for 48 h to a constant weight and then weighed. Subsamples of each soil sample were separated to determine the gravimetrical water content and soil chemical properties. Briefly, to determine the soil organic C (SOC) content, we mixed a 0.5 g soil sample, 5 ml of concentrated sulfuric acid (18.4 mol L$^{-1}$), and 5.0 ml of aqueous potassium dichromate ($K_2Cr_2O_7$) (0.8 mol L$^{-1}$) in a 100 ml test tube, then heated them in a paraffin oil pan at 190°C, keeping them boiling for 5 minutes. After cooling, the 3 drops of phenanthroline indicator were added and then the sample was titrated with ferrous ammonium sulphate (0.2 mol L$^{-1}$) until the color of the solution changed from brown to purple to dark green (Nelson and Sommers, 1982; Chen et al., 2008; Edwards et al., 2013). The soil ammonium-nitrogen (N) ($NH_4^+$-N) concentration and the nitrate-N ($NO_3^-$-N) concentration were extracted with a potassium chloride (KCl) solution and measured using a flow injection analyzer (SEAL Auto Analyzer 3; SEAL Analytical, Inc., Mequon, WI, USA) (Liu et al. 2014). Soil samples (0–10 cm in depth) from each collar were oven-dried at 105°C for at least 48 h and weighed to determine the SWC. The soil microbial biomass C (MBC) and microbial biomass N (MBN) were measured using the chloroform-fumigation extraction method and calculated by subtracting extractable C and N contents in the unfumigated samples from those in the fumigated samples (Liu et al., 2014; Rinnan et al., 2009). All extracts were stored at 4°C until further testing commenced.

## 2.6 Statistical analysis

All statistical analyses were performed using IBM SPSS Statistics 21.0 (IBM, Armonk, NY, USA). All the data were normal as tested by the Shapiro-Wilk method. A one-way analysis of variation (ANOVA) with LSD multiple range tests was conducted to test the statistical significance of the differences in the mean values of the soil temperature, soil moisture, $R_s$, belowground biomass, SOC, $NH_4^+$-N and $NO_3^-$-N concentrations, and MBC and MBN concentrations at depths of 0 to 10 cm among the different treatments. A linear regression analysis was also used to test the relationship between the SWC and $R_s$. The relationship between $R_s$ and the soil temperature in each treatment was tested with an exponential function.

We used $Q_{10}$ to express the temperature sensitivity of $R_s$ and calculated it according to the following equations:

$$R_s = ae^{bT_s} \tag{1}$$
$$Q_{10} = e^{10b} \tag{2}$$

Here, $T_s$ is the soil temperature, $a$ refers to the intercept of $R_s$ when the soil temperature is 0°C, and $b$ is the temperature coefficient reflecting the temperature sensitivity of $R_s$ and is used to calculate $Q_{10}$ (Lloyd and Taylor, 1994; Luo et al., 2001; Shen et al., 2015).

The relationship between $R_s$ and the SWC was further conducted to fit the Gompertz function, a sigmoid function (Gompertz, 1825; Yin et al., 2003), which could express that the linear increase is rapid followed by a leveling off:

$$R_s = a * e^{-b*(exp(-k*SWC))} \tag{3}$$

Here, a is an asymptote; the SWC halfway point of a/2 equals -ln(ln(2)/b)/c. The turning point of the maximum rate of $R_s$ increase equals ak/e when the SWC equals ln(b)/k. Thus, from the sigmoid function curve, the thresholds of the changes in $R_s$ with increasing SWC can be obtained from the Gompertz function (Gompertz, 1825; Yin et al., 2003).

A non-linear regression model was used to fit the relationship of $R_s$ with both soil temperature and soil moisture (Savage et al., 2009):

$$R_s = (R_{ref} * Q_{10}^{(T_s-10)/10}) * \beta^{(SWC_{0PT} - SWC)^2} \tag{4}$$

where $T_s$ is the soil temperature at a soil depth of 5 cm, $R_{ref}$ is $R_s$ at 10°C and $Q_{10}$ is a unitless expression in $R_s$ for each increase in 10°C. SWC is water content in 0 to 20 cm soil depth, $SWC_{0PT}$ is the optimal water content and $\beta$ is a parameter

modifying the shape of the quadratic fit.
Following the key factors selected by the stepwise regression method, a path
analysis was used to examine the primary components directly and indirectly
affecting $R_s$ by integrating both the stepwise linear regression module and Pearson
correlation analyses (Gefen et al., 2000). The statistical significances were set at $P$
$< 0.05$ for all tests, unless otherwise indicated.

## 3 Results

### 3.1 Warming effects on belowground characteristics

The soil temperatures at a soil depth of 5 cm in the warmed plots were much higher
than those in the control plots (Figure 1). During growing season, the mean soil
temperatures in the control, the moderately and acutely warmed plots were 21.9°C
(±0.13 SE), 24.5°C (±0.15), and 25.0°C (±0.18), respectively. The moderately and
acutely warmed plots were respectively increased by 2.6°C ($P < 0.001$) and 3.1°C
($P < 0.001$) compared to those in the control plots. The SWC in the moderately and
acutely warmed plots (0–20 cm soil profile, defined as ratios of water volume and
soil volume) were significantly reduced ($P < 0.001$) compared to those in the
control plots (Figure 1), indicating that warming led to marked declines in the SWC,
consequently enhancing drought stress. On August 18, 19, 20 and 21, which were
the dates that we measured $R_s$, the daily soil temperatures in the moderately and
acutely warmed plots were around 3°C and 4°C higher than those in the control
plots, respectively. All belowground variables (belowground biomass, soil N and
microbial characteristics) were not significantly altered by warming regimes at the
site of this experiment (Supplementary Table S1; $P > 0.05$). However, the organic
soil carbon content tended to decrease with long-term climatic warming.

### 3.2 Watering pulse effects on $R_s$

Soil respiration significantly increased with SWC both linearly ($R^2 = 0.83$; $P <$
$0.01$) and quadratically ($R^2 = 0.88$; $P < 0.01$, Figure 2A). Moreover, the Gompertz
function was well fitted to their relationship ($R^2 = 0.87$; RMSE = 4.88) (Figure 2B).
From the Gompertz functional curve, the $R_s$ asymptote value, as an estimated
maximum, was 3.76 $\mu \cdot mol \cdot m^{-2} \cdot s^{-1}$ when the optimal SWC was 22.85%. In the
watering plots, an exponential function was well fitted to the relationship between
soil respiration and the soil temperatures ($R^2 = 0.31$; $P < 0.01$), with a temperature
sensitivity ($Q_{10}$) of 1.69. However, the exponential function was not well fitted in
the control plots (Figure 3A).

### 3.3 Effects of warming regimes on $R_s$

Warming regimes resulted in marked declines in $R_s$. Whereas no difference in $R_s$
was observed in July, during August average $R_s$ values were 1.57, 1.06, and 0.93
$\mu \cdot mol \cdot m^{-2} \cdot s^{-1}$ in the control, moderately warmed and acutely warmed plots,
respectively, indicating that warming regimes resulted in marked declines (Figure
4). Changes in $R_s$ differed significantly between the control and both warmed plots
($P < 0.01$), while the $R_s$ in the two warmed plots did not significantly differ from
each other ($P = 0.45$). The relationships between the $R_s$ and soil temperature of
each treatment were well fitted by the exponential equations ($P < 0.05$) (Figure
3B). The $Q_{10}$ values were 1.88, 2.12 and 1.58 in the temperature controlled,
moderate and acute warming treatments, respectively (Figure 3B). It indicated that
$R_s$ increases exponentially with temperature in watered plots but was lower and
insensitive to temperature in the control plots (Figure 3A); and that long-term
warming rather than temporary high temperature reduced $R_s$, despite having a
positive relationship with soil temperature (Figure 3B, 4).
**3.4 Interactive effects on $R_s$ from soil temperature and soil water content**
Across all watering and warming treatments, generally, a high temperature led to
an increase in $R_s$ under ample soil moisture, whereas $R_s$ was limited under a soil
water deficit. As shown in Figure 5, A non-linear regression model (equation 4)
was well fitted to the relationship of $R_s$ with both soil temperature and soil moisture
in the control plots ($R^2 = 0.40$, RMSE = 0.60). Based on the function $R_s =$
$(0.733*1.796^{(Ts-10)/10})*\beta^{(0.229-SWC)^2}$, the key parameters were obtained: $R_{ref}$, a $R_s$ at
10°C, was 0.73 $\mu \cdot mol \cdot m^{-2} \cdot s^{-1}$; $Q_{10}$, a unitless expression in $R_s$ for each increase in
10°C, was 1.80; and $\beta$, a parameter modifying the shape of the quadratic fit, was
0.001 (Figure 5).
**3.5 Effects of multiple factors on $R_s$: a path analysis**
Based on a stepwise regression analysis of the relationships between the $R_s$ and
multiple factors, four key factors were screened: soil temperature, soil moisture,
belowground biomass and SOC. Their effects on $R_s$ were further determined by a
path analysis. The results showed that soil moisture and soil temperature were two
major direct factors controlling $R_s$ (the two direct path coefficients were 0.72 and
0.55, respectively). SOC had the highest indirect effect on $R_s$ (the indirect path
coefficient was 0.57). Soil moisture highly correlated with $R_s$ (R = 0.78, $P < 0.01$;
Supplementary Table S2, Figure 6), indicating again that the soil water status may
impose the greatest effect on the carbon release from soil in the desert grassland.
# 4. Discussion
## 4.1 Warming effects on $R_s$
Previous studies have shown positive $R_s$ responses to increased soil temperatures
below a critical high temperature (e.g., Carey et al., 2016; Drewitt et al., 2002;
Gaumont-Guay et al., 2006; Meyer et al., 2018). However, in the current study site,
the climatic warming finally reduced the average $R_s$ by 32.5% and 40.8% under
long-term versus short-term climatic warming conditions in the desert dryland,

respectively, which chiefly confirmed our first hypothesis. In a semiarid grassland on the Loess Plateau of China, the total $R_s$ was also constrained substantially by a field manipulative experiment (Fang et al., 2018). This result may have been caused by the following factors. First, high temperatures may cause thermal stress on microbes and subsequently reduce microbial respiration (i.e., heterotrophic, $R_h$, Chang et al., 2012; Dacal et al., 2019). For instance, in an alpine steppe on the Tibetan Plateau, microbial respiration was significantly reduced when the temperature rose to 30°C (Chang et al., 2012). Second, in the desert grassland, where water is often limited, the SWC becomes the primary factor affecting $R_s$ (Supplementary Table S2; Figure 6), while warming can cause greater evapotranspiration, consequently lessening soil moisture (Figure 1), and finally reducing $R_s$ (Munson et al., 2009; Wan et al., 2007; Yan et al., 2013). The decreases in average $R_s$ with warming implicate that the positive feedback loop could be weakened with length or intensity of warming.

Total respiration ($R_s$) [the sum of root (autotrophic, $R_a$) and $R_h$ respiration–the former accounting for *c.* 22 % of the total $R_s$ in the ecosystem, Liu et al., 2016] may acclimatize to warming within an appropriate range of temperature change at an ample soil moisture; however, it decreases with increasing temperatures above an optimum level. The mechanisms may include: within an appropriate range of temperature change at an ample soil moisture, climatic warming can enhance both plant root (Luo et al., 2001; Liu et al., 2016) and microbial activities (Tuker et al., 2014), leading to increases in both $R_a$ and $R_h$ , and consequently the $R_s$ (Luo et al., 2001; Tuker et al., 2014; Xu et al., 2019). However, when warming continues or with increasing temperatures above an optimum level, the root growth can be constrained, directly reducing $R_a$ (Carey et al., 2017; Liu et al., 2016; Luo et al., 2001; Wan et al., 2007); and the limitation to microbial activities may also occur (Tucker et al., 2013; Yu et al., 2018), decreasing the $R_h$ (Bérard et al., 2011; Tucker et al., 2013; Bérard et al., 2015; Romero-Olivares et al., 2017). In addition, decreases in soil enzyme pools and its activity under warming may also contribute to a reduction in $R_h$ (e.g., Alvarez et al., 2018). Further, $R_s$ decreases with warming under water deficit (Moyano et al., 2013; Wang et al., 2014; and see below). Together, the declines in both root and microbial respirations finally reduce the $R_s$. Nevertheless, the drastic declines in $R_s$ under both long-term and short-term climatic warming regimes in the desert dryland ecosystem may be driven by multiple factors, including the ecosystem type, time and soil features (Liu et al., 2016; Wan et al., 2007; Meyer et al., 2018; Thakur et al., 2019). It implies that the effects of multiple factors should be considered in assessing the carbon balance between ecosystem and atmosphere.

**4.2 Interactive effect of soil water status and temperature**

As stated above, in an arid ecosystem, soil water deficit is a primary factor
inhibiting soil carbon release (Supplementary Table S2; Figure 6; Liu et al., 2016;
Munson et al., 2009; Yan et al., 2013). Thus, $R_s$ linearly increases with increasing
soil moisture. However, it could be leveled off or decreased when soil moisture
exceeds an optimal level for the soil carbon release (Huxman et al., 2004; Moyano
et al., 2013; Wang et al., 2014). Thus, the relationship between $R_s$ and SWC may
be well fitted to the Gompertz functional curve model, a sigmoid function
(Gompertz, 1825; Yin et al., 2003), which can be confirmed by the present results
in the native arid desert ecosystem (Figure 2). The mechanisms mainly are: an
increase in SWC may rapidly increase microbial activities (Cable et al., 2008;
Meisner et al., 2015; Wu & Lee, 2011), and enhance root growth (Xu et al., 2014),
leading to a linear increase in $R_s$. However, when soil moisture reaches an ample
level, microbial activities may also reach a maximum where the limiting effects of
substrate occur (Skopp et al., 1990), finally maintaining a stable change in $R_h$.
Similar response to watering appears for root growth (Xu et al., 2014), and also
similarly leading to a stable change in $R_a$. Thus, $R_s$ can be leveled off at an increased
and stable level. Moreover, the decrease in $R_s$ at a saturated SWC level may be
ascribed to inhibitions of both root systems and microbial activities under the
anaerobic environment (Drew 1997; Huxman et al., 2004; Kwon et al., 2019;
Sánchez-Rodríguez et al. 2019; Yan et al., 2018). The model concerning the
relationship $R_s$ with a broad range of SWC is helpful to assess and predict the
dynamics in soil carbon release in natural arid ecosystems.
As indicated by Tucker and Reed (2016), soil water deficit can shrink the $R_s$
itself and its response to temperature, suggesting the changes in $R_s$ may be
determined simultaneously by both soil temperature and water status (Janssens et
al., 2001; Yan et al., 2013; Sierra et al., 2015). Moreover, in the present experiment,
the interactive effects of both factors were tested based on the relationship of $R_s$
with both soil temperature and soil moisture in a non-linear regression model
(Savage et al., 2009). The model utilized was well fitted but marginally so ($R^2 =$
0.40, RMSE = 0.596; Figure 5), indicating that both the soil temperature and soil
water content coordinated the changes in $R_s$. However, this interaction may also be
affected simultaneously by other abiotic and biotic factors, such as soil nutrition
availability and soil microbe activity (e.g., Camenzind et al., 2018; Karhu et al.,
2014; Thakur et al., 2019; Zhang et al., 2014).

**4.3 Key factors and the influence path**
As noted above, $R_s$ is affected by several abiotic and biotic factors. The current
results showed that soil moisture and soil temperature were two major direct
factors, and SOC only was an indirect factor controlling $R_s$ (Supplementary Table
S2, Figure 6). Importantly, soil moisture, with both the highest direct path
coefficients (0.7) and correlation coefficient (0.8) for $R_s$, may become the most

important factor affecting $R_s$ in this desert steppe. These findings agree with the previous results: improved soil water status had a significantly positive effect on $R_s$ (e.g., Chen et al., 2008; Liu et al., 2016; Xu et al., 2016). Furthermore, the soil moisture condition can mediate the relationship between soil temperature and $R_s$, thus affecting its temperature sensitivity; SWC becomes the main factor controlling $R_s$, especially in arid ecosystems, such as desert steppes, where the available soil water is limited (Conant et al., 2000; Curiel et al., 2003; Fa et al., 2018; Jassal et al., 2008; Roby et al., 2019). Thus, under both the long-term and short-term climatic warming regimes, soil moisture could modulate the response of $R_s$ to warming. The changes in $R_s$ might be driven by both soil temperature and soil moisture as two key factors, and SOC as an indirect factor, thus mostly confirming our second hypothesis. The findings again implicate that multiple factors together coordinate $R_s$, and provide new insight into how to control soil carbon release in arid ecosystems. The models on the $R_s$ changes should consider multiple-factor effects of soil carbon dynamics when assessing and predicting carbon cycle, and its climate feedback.

**4.4 Warming effects on the variables belowground**

Elevated temperature has been shown to increase or decrease root productivity and biomass, depending on experimental sites and vegetation types (Bai et al., 2010; Fan et al., 2009; Litton and Giardina, 2008; Wan et al., 2004). The decreased availability of soil nutrients apparently limits root growth, finally inducing root mortality and weakening responses to the elevated temperature (Eissenstat et al., 2000; Johnson et al., 2006; Wan et al., 2004; Zhang et al., 2014). In our experiment, no significantly different changes occurred in either soil $NH_4^+$-N or $NO_3^-$-N concentrations among the three treatments (Supplementary Table S1), and these might be linked to the non-significant response of belowground biomass to increasing temperature. Microbial biomass and its activities in soil depend on the root biomass, SWC and soil N conditions (Liu et al., 2014; Rinnan et al., 2007; Zhang et al., 2008; Zhang et al., 2014). Warming regimes had no significant effects on either MBC or MBN in the current study (Supplementary Table S1), which might be due to the lack of any difference in the changes in basic soil nutrition status, such as the N conditions, among the three warming treatments. This result is consistent with that of Zhang et al. (2005) and Liu et al. (2015). Moreover, in the present study, SOC concentrations were not significantly affected by climatic warming (Supplementary Table S1), which is inconsistent with the findings of previous studies (Jobbágy and Jackson, 2000; Prietzel et al., 2016). However, there might be a decreasing trend evident with long-term warming. For instance, Crowther et al. (2016) reported a loss of approximately $30 \pm 30$ Pg of C in the upper soil horizons at 1℃ warming in global soil C stocks and projected a loss of $203 \pm 161$ Pg of C under 1℃ of warming over 35 years. The C losses from soil

moving into the atmosphere may result in positive feedback regarding global warming (Bradford et al., 2016; Dacal et al., 2019; Jenkinson et al., 1991; Liu et al., 2016). However, SOC exerted an indirect effect via a path analysis (Figure 6). For this difference, therefore, more evidence needs to be provided to address the issue (Xu et al., 2019).

In conclusion, we determined the responses of $R_s$ to field experimental long-term versus short-term climatic warming and watering pulses in a desert steppe ecosystem. We found the following: i) both long- and short-term warming significantly reduced $R_s$ during the peak growth season; ii) soil moisture was the main factor controlling $R_s$ in desert grassland; iii) $R_s$ was significantly and exponentially increased with soil temperature, meanwhile soil moisture condition can mediate the relationship between soil temperature and $R_s$, thus affecting its temperature sensitivity; and iv) belowground biomass, soil nutrition variables and soil microbial characteristics showed no significant changes after either long-term or short-term climatic warming. These findings may be useful to assess and predict dynamics of soil $CO_2$ fluxes, particularly the feedback of warming to climatic change, and finally optimize C management work in arid and semiarid regions under the changing climate. However, the patterns of the changes in soil C fluxes and the underlying mechanism in response to climatic change are markedly complicated at various spatial-temporal scales during growing season—from site and regional to global scales, and from daily, seasonal and yearly to decade scales––and still need to be investigated further (e.g., Ballantyne et al., 2017; Dacal et al., 2019; Meyer et al., 2018; Romero-Olivares et al., 2017).

***Data availability***. The final derived data presented in this study are available at https://doi.org/10.5281/zenodo.3546062 (Yu et al., 2019).

***Supplement***. The supplement related to this article is available online at:

***Author contributions***. ZX and GZ conceived and designed this study. HY, ZX and YS conducted this experiment and analysed the data. All authors wrote and proofread this manuscript.

***Competing interests.*** The authors declare that they have no conflict of interest.

***Acknowledgements.*** This research was jointly funded by National Natural Science Foundation of China (31661143028, 41775108), and the Special Fund for Meteorological Scientific Research in the Public Interest (GYHY201506001-3). We greatly thank Feng Zhang, Yuhui Wang, Bingrui Jia, Hui Wang, Minzheng Wang, He Song for their loyal help during the present study. The authors also greatly appreciate Dr. De Kauwe and the reviewers for their constructive comments.

491

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

**Figure legends**

Figure 1. Effects of warming on the soil temperature and soil moisture during the growth peak in 2014 (Mean ± SE). Mean daily values were presented (n = 120). The mean values with the same lowercase letters on the SE bars are not different at $P < 0.05$ according to LSD multiple range tests ($P$ values and F ratios are shown inside).

Figure 2. Relationship between $R_s$ and soil water content based on a linear (blue line) and a quadratic (black line) functional model (A), and Gompertz functional model (B). Close and open circles denote the data in 2014 and 2017, respectively. The close red circles indicate data used for the linear $R_s$ response to SWC at low levels. The one open triangle may be an outlier point due to some errors, but it does not notably affect the functional fitting when removing it (ref. Figure S2). Based on Gompertz functional curve, the $R_s$ asymptote value, as an estimated maximum, is 3.76 $\mu \cdot mol \cdot m^{-2} \cdot s^{-1}$ when the optimal SWC is 22.85% [The red line denotes the initial $R_s$ response to SWC; the blue line denotes $R_s$ = constant value of the maximum estimated by the asymptote value; and the intersection of the two lines represents a point (the blue arrow) at which $R_s$ levelled off]. Note, we measured the $R_s$ during 9:00-10:00 in these cloudless days with calm/gentle wind in order to maintain other environmental factors such as soil temperature and radiation to relatively stable and constant. The data were collected in the plots of watering treatments (n = 92)..

Figure 3. The relationships between soil respiration and soil temperature under both watering (n = 23-25, A), and warming treatments (n=28-33, B) (Mean ± SE).

Figure 4. Effects of warming regimes on average soil respiration in 2014 (mean ± SE), the mean values with the same lowercase letters on the SE bars are not different at $P < 0.05$ according to LSD multiple range tests ($P$ values and F ratios are shown inside).

Figure 5. An interactive relationship of soil respiration with both soil temperature (Ts) and soil water content (SWC) based on a nonlinear mixed model ($R_s = (0.733*1.796^{(Ts-10)/10})*\beta^{(0.229-SWC)^2}$). The data were used in control plots in the warming experiment. The optimal SWC of 0.229 was estimated by the Gompertz functional curve (see Figure 2B).

Figure 6. A diagram of the effects of key environmental factors on soil respiration and their relationships. Blue double-headed arrows represent the relationships between the key environmental factors, data on the arrows are correlation coefficients. Black arrows represent the relationships between soil respiration and the key environmental factors, data on the arrows are correlation coefficients (bold) and direct path coefficients (italic), respectively. *, $P < 0.05$; **, $P < 0.01$, n = 12. For other details, see Supplementary Table S2.

Supplementary Figure S1. Long-term air temperature (A) and total annual

precipitation (B) records from 1955 to 2014 in the experiment site in the desert
steppe ecosystem, Damao Banner, Nei Mongol, China.
Supplementary Figure S2. Relationship between $R_s$ and soil water content based
on a linear (black line) and a quadratic (dotted line) functional model (A), and
Gompertz functional model (B). Close and open circles denote the data in 2014
and 2017, respectively. The close red circles indicate data used for the initial $R_s$
response to SWC. The functional fitting does not substantially affect despite a
slight improvement with greater $R^2$ values when the outlier point was removed (ref.
Figure 2). Note, we measured the $R_s$ during 9:00-10:00 in the cloudless days with
calm/gentle wind in order to maintain other environmental factors such as soil
temperature and radiation to relatively stable and constant (n = 91).

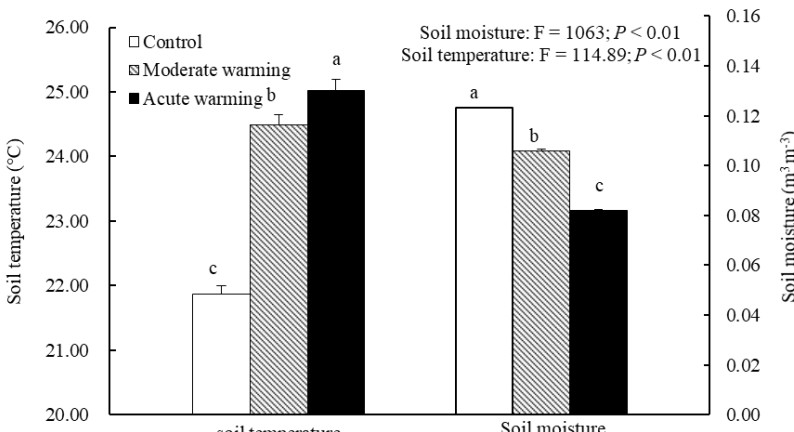

**Figure 1.**    Effects of warming on the soil temperature and soil moisture during the growth
peak in 2014 (Mean ± SE). Mean daily values were presented (n = 120). The mean values with
the same lowercase letters on the SE bars are not different at $P < 0.05$ according to LSD
multiple range tests ($P$ values and F ratios are shown inside).

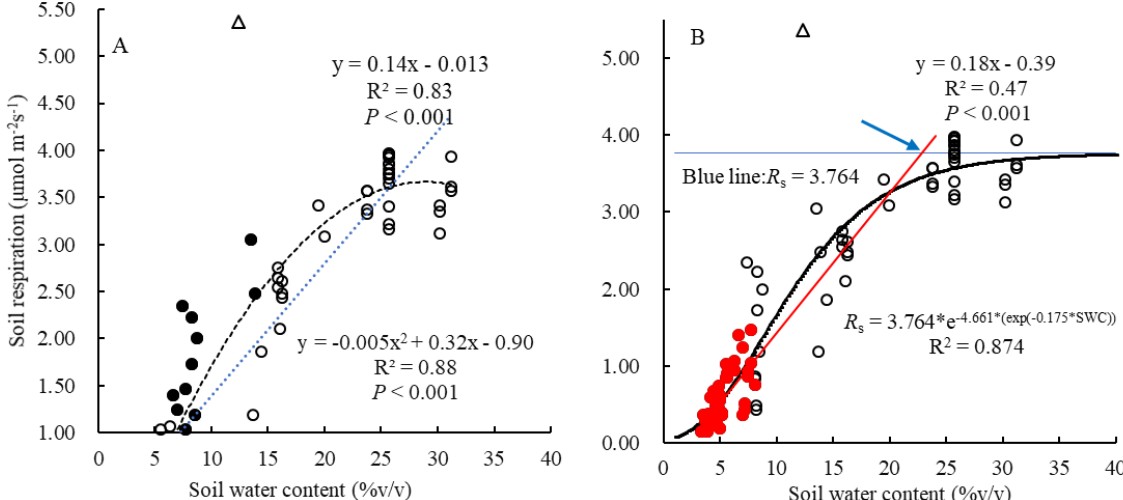

**Figure 2.** Relationship between $R_s$ and soil water content based on a linear (blue line) and a quadratic (black line) functional model (A), and Gompertz functional model (B). Close and open circles denote the data in 2014 and 2017, respectively. The close red circles indicate data used for the linear $R_s$ response to SWC at low levels. The one open triangle may be an outlier point due to some errors, but it does not notably affect the functional fitting when removing it (ref. Figure S2). Based on Gompertz functional curve, the $R_s$ asymptote value, as an estimated maximum, is 3.76 μ·mol·m$^{-2}$·s$^{-1}$ when the optimal SWC is 22.85% [The red line denotes the initial $R_s$ response to SWC; the blue line denotes $R_s$ = constant value of the maximum estimated by the asymptote value; and the intersection of the two lines represents a point (the blue arrow) at which $R_s$ levelled off]. Note, we measured the $R_s$ during 9:00-10:00 in these cloudless days with calm/gentle wind in order to maintain other environmental factors such as soil temperature and radiation to relatively stable and constant. The data were collected in the plots of watering treatments (n = 92).

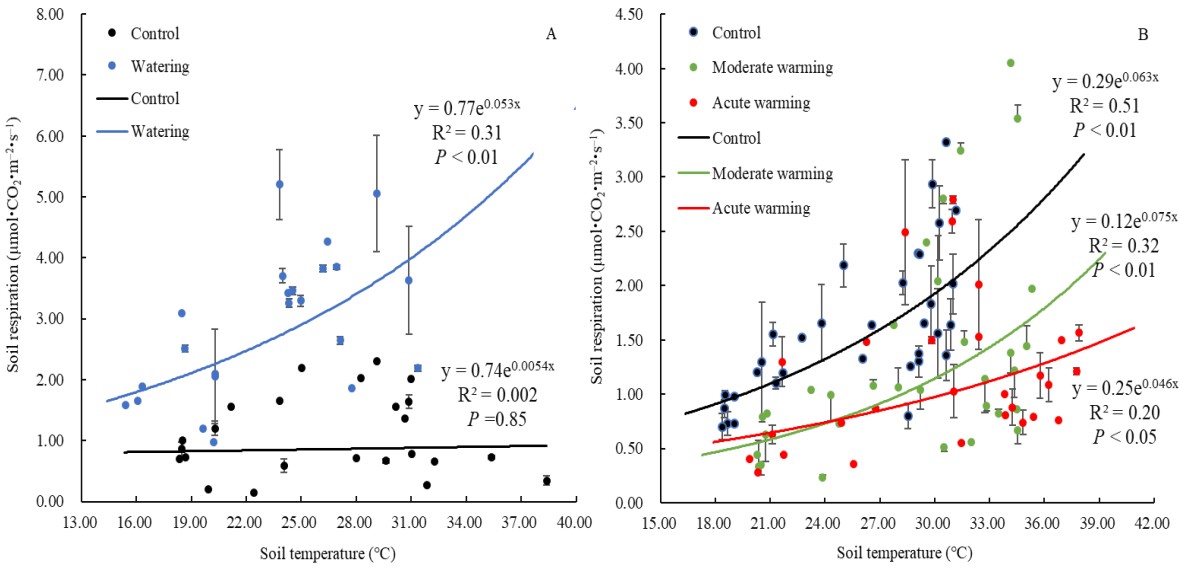

**Figure 3.** The relationships between soil respiration and soil temperature under both watering
(n = 23-25, A), and warming treatments (n=28-33, B) (Mean ± SE).

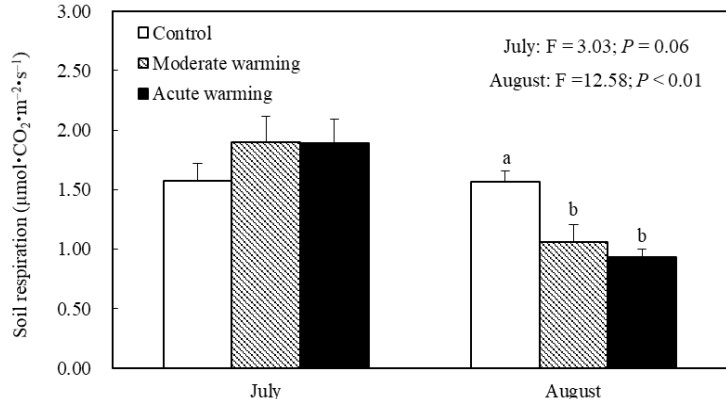

**Figure 4.** Effects of warming regimes on average soil respiration in 2014 (mean ± SE), the
mean values with the same lowercase letters on the SE bars are not different at $P < 0.05$
according to LSD multiple range tests ($P$ values and F ratios are shown inside).

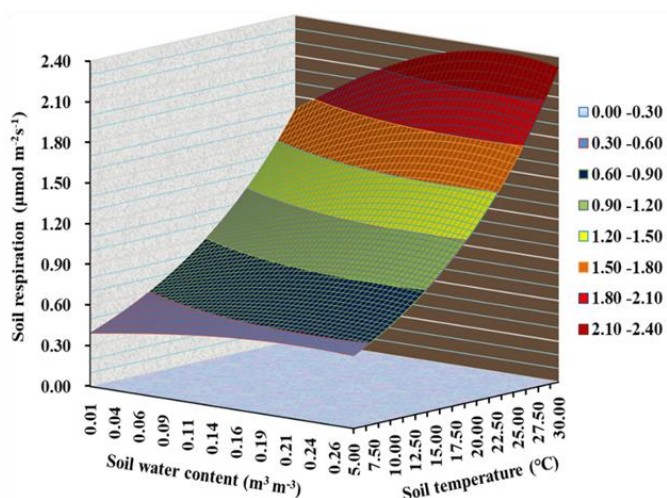

**Figure 5.** An interactive relationship of soil respiration with both soil temperature (Ts) and soil
water content (SWC) based on a nonlinear mixed model ($R_s = (0.733*1.796^{(Ts-10)/10})*\beta^{(0.229-SWC)^2}$,
B). The data were used in control plots in the warming experiment. The optimal SWC of 0.229
was estimated by the Gompertz functional curve (see Figure 2B).

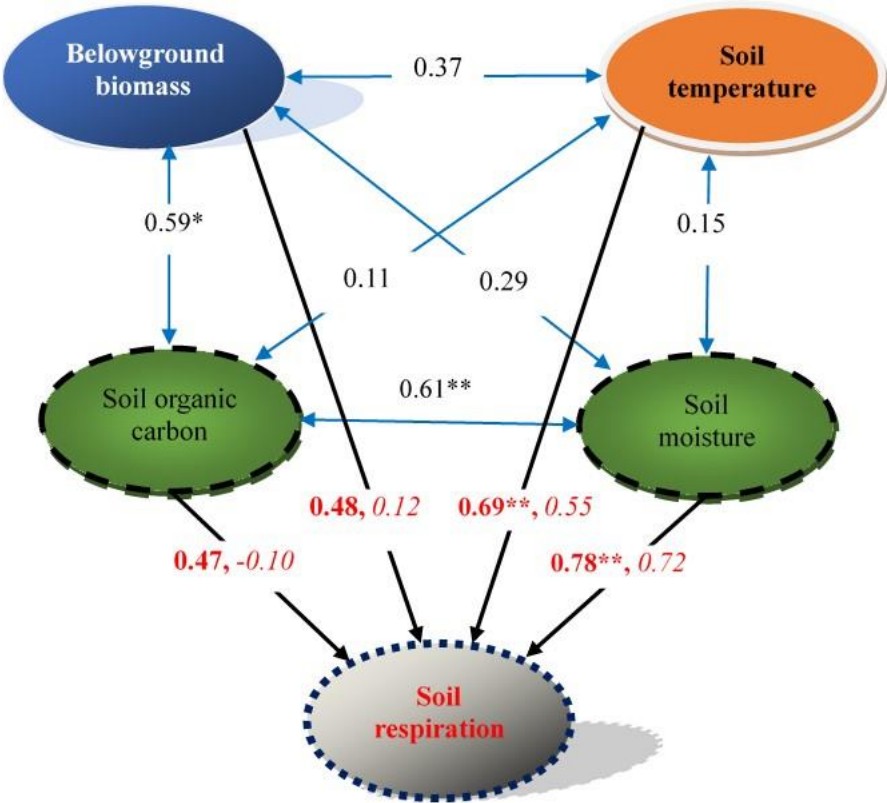

**Figure 6.** A diagram of the effects of key environmental factors on soil respiration and their relationships. Blue double-headed arrows represent the relationships between the key environmental factors, data on the arrows are correlation coefficients. Black arrows represent the relationships between soil respiration and the key environmental factors, data on the arrows are correlation coefficients (bold) and direct path coefficients (italic), respectively. *, $P < 0.05$; **, $P < 0.01$, n = 12. For other details, see Supplementary Table S2.