# Peer review of "Soil carbon release responses to long-term versus short-term climatic"

_Biogeosciences, 2019_

## Referee Comment (RC1) · Anonymous Referee #2 · 1 Oct 2019

Title: Soil carbon release responses to long-term versus short-term climatic warming in an arid ecosystem

General Comments:

This paper specifically addresses our lack of knowledge on climate change within desert grassland ecosystems. While there have been many studies within boreal and temperate ecosystems, warming experiments focused on soil microbial community function and C stock assessments in arid systems are rare. Therefore, this study fills an important gap.

The sampling design appears robust and is accompanied with a clear presentation of data. The methods for analyses are valid and informative enough for readers that might not be familiar with these techniques to understand, or where to read more, if desired. The authors further supported their hypotheses succinctly that warming reduced microbial respiration while wetting events enhanced respiration. However, my one concern is that 0.5 – 1.0 degree (Line 244) warming is not enough of a difference to define two treatments of warming (long-term, moderate and short-term, acute). Although, when moisture became limiting in August, there was a greater separation of the warming treatments which does highlight seemingly subtle differences. Perhaps a more direct addressing of the miniscule differences in the warming treatments until SWC becomes the greatest inhibitor to respiratory rates.

Overall, I find this study well conducted and clearly presented. I would recommend for publication.

Specific Comments:

Line 12: Can examples of severe impacts be included here?

Lines 16-17: Since many readers do not get past an abstract, it would be useful to list treatment pressures in parantheses following long-term (ex.), short-term, etc.

Line 19: Give the percentage of substantial water input treatment?

Line 55: Use a more updated IPCC statistics (this one is 2014)

Comment: Warming treatment pressures were only applied during the growing season (June-Aug) of each year. In many temperate ecosystems, there has been evidence of seasonal extensions. Has there been any evidence that the growing season is becoming extended (earlier springs, later falls) at this site? If so, should this warrant extending warming treatments beyond June – August?

Line 774: If outlier point does not change equations, why not remove it completely?

Technical Comments:

Figure 2 line 769: Linear line is blue and not black as stated

Line 773: Typo? What does "soil animal" refer to?

—————————————————————

---

## Referee Comment (RC2) · Anonymous Referee #3 · 25 Oct 2019

**GENERAL COMMENTS**

This study addresses an important research topic—climate change impacts on dryland soil carbon dynamics. This article presents valuable data from a field manipulation study in which the authors examined how warming and watering regimes of varying intensity and duration impact soil respiration in a desert steppe. While the study methods appear sound and the results provide strong evidence for warming-driven reductions in soil respiration, many sections in the text are unclear and need to be improved to strengthen and clarify the manuscript. The authors could modify hypothesis two into a statement that could be tested in this study and contribute to new insight on the dy-

namics of soil respiration in water-limited ecosystems. There are key findings that are not clearly reported and challenge my interpretation as a reader. Specifically, the authors should address an apparent conflict: warming decreased Rs despite the positive relationship between Rs and soil temperature. The authors should explicitly highlight the important role of soil moisture as the dominant control on Rs rates and temperature sensitivity. Lastly, the data availability statement does not appear to meet the journal's data policy requirements, and I suggest uploading data to a public repository, if possible.

SPECIFIC COMMENTS Parts of this manuscript would benefit from additional explanation. Below I provide some specific examples.

L 24-27. "This indicates that soil carbon release responses strongly depend on the duration and magnitude of climatic warming, which may be driven by SWC and soil temperature." This is unclear. Please explain how SWC and soil temperature influence soil respiration, and then perhaps infer how those relationships have implications for climatic warming impacts on soil carbon dynamics.

L 55-59: An explanation of why low precipitation and biomass enhances vulnerability would strengthen the authors' claim that deserts are sensitive to climate change.

L 60-66: This section shows that temperature and moisture are well-known controls on Rs. However, this conflicts with the previous claim (L 43-47) that Rs responses to biotic and abiotic factors are poorly understood. Can this apparent contradiction be addressed in a way that makes a stronger case for this study? E.g. whereas soil moisture and temperature are well-known controls on Rs, it is not well known how soil moisture modulates the response of Rs to changes in the duration and intensity of warming.

L 84: Please elaborate on "undefined" since many studies have reported Rs pulses after water inputs (Huxman et al., 2004; Sponseller, 2007).

Huxman, Travis E., et al. "Precipitation pulses and carbon fluxes in semiarid and arid ecosystems." Oecologia 141.2 (2004): 254-268.

Sponseller, Ryan A. "Precipitation pulses and soil CO2 flux in a Sonoran Desert ecosystem." Global Change Biology 13.2 (2007): 426-436.

L 86-88. This argument would be stronger if the authors explained why a long-term study (4 years) might yield insights undetected in previous two-year studies. Why do the authors expect to find something new?

L 88-89: Unclear. Please elaborate.

L 97-98: The introduction section already provides evidence in support of H2. In its current form, it is not clear why it is worth testing H2 in this study. How could H2 be modified into a hypothesis that could be tested in this study and contribute to new insight on the dynamics of soil respiration in water-limited ecosystems?

Results

3.1. Warming effects on soil features L 251-254: According to the Supplementary Table S1, belowground biomass is 11.5 units for the Acutely Warmed treatment. Is this a typo? It is considerably higher than the BB reported for other treatments.

3.2: It is unclear why this is section is titled "Watering pulse effects on Rs." Does this section refer to data collected only after watering? Or does the section report findings from all measurement dates? Figure 2: Please explain the data source – do the data represent the control or warmed treatments? Also, is it necessary to show the linear and quadratic fits? Are these pieces of information reported or used to make inferences?

Figure 3A. This figure presents information that is critical for the authors' conclusion. It provides evidence for why Rs was lower in warmed treatments, despite having a positive relationship with soil temperature. I suggest leading Section 3.2 or 3.3 with a strong statement describing the relationship between Rs, temperature, and moisture.

For example, soil respiration increased exponentially with temperature in watered plots but was lower and insensitive to temperature in the control plots.

L 771: Unclear. What is the initial Rs response to SWC? What do the other points represent?

Section 3.3 Suggest leading with conclusive evidence. For example, "Warming regimes resulted in marked declines in Rs. Whereas no difference in Rs was observed in July, during August average Rs values were x, y, z for the control, moderately warmed, and acutely warmed treatments, respectively."

Section 3.4 needs a figure reference.

Discussion

This section should explain why Rs decreased in warmed plots despite having a positive relationship with soil temperature.

L 319-322: Unclear how Rs can acclimate to warming but also decrease. Please explain the mechanism. Is the acclimation referring to changes in microbial respiration? Are net reductions in Rs driven by temperature-stress impacts on plant and root activity?

L358-362: Consider citing previous studies documenting that the temperature response of Rs is conditional on moisture (Roby et al., 2019; Conant et al., 2000)

Roby, M. C., Scott, R. L., Barron-Gafford, G. A., Hamerlynck, E. P., & Moore, D. J. (2019). Environmental and Vegetative Controls on Soil CO2 Efflux in Three Semiarid Ecosystems. Soil Systems, 3(1), 6.

Conant, Richard T., Jeffrey M. Klopatek, and Carole C. Klopatek. "Environmental factors controlling soil respiration in three semiarid ecosystems." Soil Science Society of America Journal 64.1 (2000): 383-390.

TECHNICAL COMMENTS

L22: Features is unclear.

L 143: What are the units of soil moisture?

L 227: Please provide depth of soil temperature measurements.

L 199: First mention of SWC; please define or introduce this acronym in section 2.3

L 126. Unclear. Is 1 m the wavelength of radiation or dimension of the heater?

L117-119: Suggest using concise and consistent treatment names. E.g. control, long term moderate warming, short-term acute warming.

L 283: "Mode" typo.

L 283: Please provide equation number.

L 238: Suggest different word for features

L 241-243: Suggest reporting an error estimate instead of range

L 246: Define v/v

Throughout: Be consistent with significant figures (L 264 : R2 = 0.31 vs. L 284: R2 = 0.404

References

Conant, Richard T., Jeffrey M. Klopatek, and Carole C. Klopatek. "Environmental factors controlling soil respiration in three semiarid ecosystems." Soil Science Society of America Journal 64.1 (2000): 383-390.

Huxman, Travis E., et al. "Precipitation pulses and carbon fluxes in semiarid and arid ecosystems." Oecologia 141.2 (2004): 254-268.

Roby, M. C., Scott, R. L., Barron-Gafford, G. A., Hamerlynck, E. P., & Moore, D. J. (2019). Environmental and Vegetative Controls on Soil CO2 Efflux in Three Semiarid Ecosystems. Soil Systems, 3(1), 6.

Sponseller, Ryan A. "Precipitation pulses and soil CO2 flux in a Sonoran Desert ecosystem." Global Change Biology 13.2 (2007): 426-436.

---

## Referee Comment (RC3) · Anonymous Referee #4 · 30 Oct 2019

The paper describes a four year warming and wetting experiment in a desert steppe in Northern China. The introduction gives a good overview of the latest and more established scientific insights and the authors did a thorough measurement campaign. I particularly appreciate how much work went into the various additional belowground measurements. Given the limited number of such experiments for this ecosystem type, this work is certainly of interest to readers of Biogeosciences. Overall, the paper is well structured and written clearly, but I feel there are some elements in the text that require clarification or some more in-depth information. If the authors manage to improve these elements I would recommend the paper for publication.

[Figure]

Major comments:

My first major comment is directly about the abstract. There is a seemingly counter-intuitive message there that confused me when reading it: Long-term warming reduced Rs by 32.5 percent (line 18). Yet, long term climatic warming decreased SOC (line 24)? While this is certainly possible, it is not directly what one would expect. Was this reduction in SOC caused by an initial spike in Rs at the beginning of the experiment? The lower SOC content could then also contribute to decreasing respiration due to reduced availability of substrate to microbes to decompose. Yet, the authors mainly talk about the moisture effect and how low soil moisture decreased Rs. The mentioned decline in SOC from the abstract is not presented in the results and discussion. Actually, the authors state that "in the present study, SOC concentrations were not significantly affected by climatic warming" and then later write "although SOC might be expected to decrease with long-term climatic warming"(conclusion iv). I do not understand how such a strong statement can be made in the abstract when the results and discussion show otherwise and even contradict one another. Given the high number of people that generally do not read beyond the abstract, my suggestion is to 1) rewrite this part of the abstract more clearly and 2) to present the evidence to support this claim more clearly in the results and discussion.

My second major comment is about the authors' choice for the various model fittings in the statistical analysis and in particular for the Gompertz function. The authors provide limited explanation for choosing the Gompertz equation in section 4.2, line 334-337 and mention the parabolic curve function as another viable option. Indeed, in section 3.2 there is another model with a better fit: the quadratic functional model. The authors do not argue further why they still continued parameter fitting with the Gompertz curve, despite the quadratic model having a seemingly better fit (figure 2 and section 3.2). I would like to know 1) why the Gompertz function was selected and 2) how picking that curve to fit the parameters for the non-linear reression model (eq 4) affected the results compared to taking the parameters from a quadratic model fitting (sensitivity analysis)?

[Figure]

Minor comments:

- The Gompertz function (line 22): This function (and its shape) might not be a given knowledge for all readers. My suggestion is to rephrase in the abstract to "whereas the relationship between Rs and soil moisture was better fitted to a sigmoid function" and explain the Gompertz curve further in Section 2.6 (see major comment #2).

- Line 48: The desert steppe is c. 8.8 million square hm. Do the authors mean total global desert steppe area or the area in China?

- Line 74/75: I would suggest adding the more recent reference to Yan et al. 2018 here as well.

Reference:

Yan, Z., B. Bond-Lamberty, K. E. Todd-Brown, V. L. Bailey, S. Li, C. Liu, and C. Liu (2018), A moisture function of soil heterotrophic respiration that incorporates microscale processes, Nature Communications, 9(1), 2562, doi: 10.1038/s41467-018-04971-6.

---

## Referee Comment (RC4) · Anonymous Referee #4 · 30 Oct 2019

I forgot to add one minor textual edit to my review:

- line 195: Replace were with was. The analysis (of vartation) is singular and therefore requires a singular verb.

---

## Author Comment (AC1) · 2 Nov 2019

Dear Dr. Bahn: Thank you for your letter and the reviewers' comments concerning our manuscript entitled "Soil carbon release responses to long-term versus short-term climatic warming in an arid ecosystem". These valuable suggestions and comments help us greatly to improve our manuscript so that we have studied them carefully and made corrections point by point according to the constructive comments. The main changes in the revised manuscript have been highlighted using red font. Please see our point-by-point responses to the reviewers' comments the in detail as following shortly. C: the original comments; R: the responses to the comments.

---

## Author Comment (AC3) · 2 Nov 2019

C: the original comments; R: the responses to the comments.

General Comments: C: This study addresses an important research topical at climate change impacts on dryland soil carbon dynamics. This article presents valuable data from a field manipulation study in which the authors examined how warming and watering regimes of varying intensity and duration impact soil respiration in a desert steppe. While the study methods appear sound and the results provide strong evidence for warming-driven reductions in soil respiration, many sections in the text are unclear and need to be improved to strengthen and clarify the manuscript. The authors could modify hypothesis two into a statement that could be tested in this study and contribute to new insight on the dynamics of soil respiration in water-limited ecosystems. R: Thank you for the positive comments. In the new version, we have revised the manuscript to strengthen and clarify the results as kindly suggested (please see detail below). The two hypotheses have been modified accordingly (Lines 105-109 of the new version, highlighted by red words).

C: There are key findings that are not clearly reported and challenge my interpretation as a reader. Specifically, the authors should address an apparent conflict: warming decreased Rs despite the positive relationship between Rs and soil temperature. The authors should explicitly highlight the important role of soil moisture as the dominant control on Rs rates and temperature sensitivity. R: Thank you for helpful comments. Actually, that is, the persistent warming treatments decreased Rs. The positive relationship between Rs and soil temperature occurs in each plot or each treatment. The two data sets are different, the former is warming treatment effect (comparison among the treatments: long-term warming, short-term warming, and ambient as a control), and the latter is the relationship between Rs and soil temperature. Yes, the important roles of soil moisture as the dominant control on Rs rates and temperature sensitivity were highlighted in many appropriate places of the newly revised version (e.g., lines, 22-23, 99-100, 374-376).

C: Lastly, the data availability statement does not appear to meet the journal's data policy requirements, and I suggest uploading data to a public repository, if possible. R: Thank you. the "Data availability" statement has been placed as at the end of the manuscript before the acknowledgements. Data availability: data can only be accessed in the form of Excel spreadsheets via the corresponding author. If necessary, we will upload it to a public repository.

Specific comments: Parts of this manuscript would benefit from additional explanation. Below I provide some specific examples. C: L 24-27. "This indicates that soil carbon release responses strongly depend on the duration and magnitude of climatic warming,

which may be driven by SWC and soil temperature." This is unclear. Please explain how SWC and soil temperature influence soil respiration, and then perhaps infer how those relationships have implications for climatic warming impacts on soil carbon dynamics. R: Thank you, we have revised it to "This indicates that climatic warming constrains soil carbon release, which is controlled mainly by decreased soil moisture, consequently influencing soil carbon dynamics" to be clearer and more concise (Lines 21-23 of the newly revised version).

C: L 55-59: An explanation of why low precipitation and biomass enhances vulnerability would strengthen the authors' claim that deserts are sensitive to climate change. R: Many thanks. We have made it to be clearer and concise accordingly, and the explanation was also added accordingly: "For instance, water deficit and heat waves during growing season can markedly decrease plant cover and productivity in this arid ecosystem" (Lines 58-62).

C: L 60-66: This section shows that temperature and moisture are well-known controls on Rs. However, this conflicts with the previous claim (L43-47) that Rs responses to biotic and abiotic factors are poorly understood. Can this apparent contradiction be addressed in a way that makes a stronger case for this study? E.g. whereas soil moisture and temperature are well-known controls on Rs, it is not well known how soil moisture modulates the response of Rs to changes in the duration and intensity of warming. R: we have changed the expressions in both sections to be clearer and more logical (Lines 44-48; 65-67). Many thanks.

C: L 84: Please elaborate on "undefined" since many studies have reported Rs pulses after water inputs (Huxman et al., 2004; Sponseller, 2007). Huxman, Travis E., et al. "Precipitation pulses and carbon fluxes in semiarid and arid ecosystems." Oecologia 141.2 (2004): 254-268. Sponseller, Ryan A. "Precipitation pulses and soil CO2 flux in a Sonoran Desert ecosystem." Global Change Biology 13.2 (2007): 426-436. R: Thank you for the useful advice, this part has been revised accordingly and cited the two references (Lines 88-92).

C: L 86-88. This argument would be stronger if the authors explained why a long-term study (4 years) might yield insights undetected in previous two-year studies. Why do the authors expect to find something new? R: Thank you, we have important findings in previous two-year which have been published (Liu et al. Plant Soil, 2016, 400:15–27). In the current study, we expect that the long-term (four-year) warming have different effects on Rs (i.e., more profound, even reverse effects relative to previous two-year short term); and the underlying mechanism under longer term warming condition, and the role of soil water status to Rs responses to climatic warming, are also uncertain (added this explanation in the new version, lines 96-100).

C: L 88-89: Unclear. Please elaborate. R: Thanks very much, we have re-edited it to "and the underlying mechanism under longer term warming condition, and the role of soil water status to Rs responses to climatic warming, are also uncertain" (also see above).

C: L 97-98: The introduction section already provides evidence in support of H2. In its current form, it is not clear why it is worth testing H2 in this study. How could H2 be modified into a hypothesis that could be tested in this study and contribute to new insight on the dynamics of soil respiration in water-limited ecosystems? R: This H2 has been modified to "the dynamics of Rs in the water-limited ecosystem can be driven mainly by the combination of soil temperature and soil moisture, and soil moisture can modulate the response of Rs to warming". Many thanks.

C: Results 3.1. Warming effects on soil features L 251-254: According to the Supplementary Table S1, belowground biomass is 11.5 units for the Acutely Warmed treatment. Is this a typo? It is considerably higher than the BB reported for other treatments. R: Thanks for your comments, this is a mistake, it should be 1.15, and we have corrected it (Supplementary Table S1).

C: 3.2: It is unclear why this is section is titled "Watering pulse effects on Rs." Does this section refer to data collected only after watering? Or does the section report findings

from all measurement dates? R: We have two experiments: one is the warming experiment which included three treatments: control, long-term moderate warming, and short-term acute warming. The other is the watering pulse treatments which included control and watering treatment. Yes, this section referred to data collected in the plots of watering treatments.

C: Figure 2: Please explain the data source – do the data represent the control or warmed treatments? Also, is it necessary to show the linear and quadratic fits? Are these pieces of information reported or used to make inferences? R: This section mainly focused on the relationship between Rs and soil water content. The data were collected in the plots of watering treatments (added in the figure 2 legend of the new version), and were used to determine the relationship between Rs and soil water content in dessert steppe. Here, we focused on the comparisons between the linear, quadratic, and Gompertz functional models. Thus, the information used could be useful. Many thanks for the kind comments.

C: Figure 3A. This figure presents information that is critical for the authors' conclusion. It provides evidence for why Rs was lower in warmed treatments, despite having a positive relationship with soil temperature. I suggest leading Section 3.2 or 3.3 with a strong statement describing the relationship between Rs, temperature, and moisture. For example, soil respiration increased exponentially with temperature in watered plots but was lower and insensitive to temperature in the control plots. R: Thanks for your useful advice, it has been revised accordingly in lines 287-289.

C: L 771: Unclear. What is the initial Rs response to SWC? What do the other points represent? R: It should be linear Rs response to SWC at low levels. This is Gompertz functional model features: for the all points, with SWC increasing, Rs linearly increased sharply, then reaching a maximum value, and levelling off. Thanks.

C: Section 3.3 Suggest leading with conclusive evidence. For example, "Warming regimes resulted in marked declines in Rs. Whereas no difference in Rs was observed in July, during August average Rs values were x, y, z for the control, moderately warmed, and acutely warmed treatments, respectively." R: Thanks, it has been done in lines 278-281.

C: Section 3.4 needs a figure reference. R: The reference figure is figure 5, and was added (Line 296).

C: This section should explain why Rs decreased in warmed plots despite having a positive relationship with soil temperature. R: They are different two terms: Rs in warmed plots were the values averaged in the warming treatments, whereas Rs values used for the relationship with soil temperature are the data in each plots or each treatments; and particularly, the soil temperature data used for the relationship Rs and soil temperature are the values when the Rs were measured simultaneously. They two are matching values each other. Thus, long-term warming rather than temporary high temperature reduced Rs, despite having a positive relationship with soil temperature (added, lines 289-291).

C: L 319-322: Unclear how Rs can acclimate to warming but also decrease. Please explain the mechanism. Is the acclimation referring to changes in microbial respiration? Are net reductions in Rs driven by temperature-stress impacts on plant and root activity? R: Rs can acclimate to warming but also decrease, that may because the soil moisture levels differs: Rs can acclimate to warming at an ample soil moisture; whereas it decreases under water deficit (added lines 333-335). Yes, the Rs includes microbial respiration, but the microbial respiration is not separated from whole Rs in the current study. The net reductions in Rs could link to temperature-stress impacts on plant and root activity (e.g., Liu et al., 2016; Luo et al., 2001). Nevertheless, the underlying mechanism needs to be explored further. Many thanks for the valuable comments.

C: L358-362: Consider citing previous studies documenting that the temperature response of Rs is conditional on moisture (Roby et al., 2019; Conant et al., 2000). Roby,

M. C., Scott, R. L., Barron-Gafford, G. A., Hamerlynck, E. P., Moore, D. J. (2019). Environmental and Vegetative Controls on Soil CO2 Efflux in Three Semiarid Ecosystems. Soil Systems, 3(1), 6. Conant, Richard T., Jeffrey M. Klopatek, and Carole C. Klopatek. "Environmental factors controlling soil respiration in three semiarid ecosystems." Soil Science Society of America Journal 64.1 (2000): 383-390. R: Thank you, we have cited them already in lines 377-378, and added in the reference list.

TECHNICAL COMMENTS C: L22: Features is unclear. R: This has been changed it to "The belowground biomass, soil nutrition, and microbial biomass" to detail these soil variables in line 26 of the new revision.

C: L 143: What are the units of soil moisture? R: It has been added (a ratio: v/v).

C: L 227: Please provide depth of soil temperature measurements. R: We have provided it in line 237.

C: L 199: First mention of SWC; please define or introduce this acronym in section 2.3 R: the SWC whole name has been added in line 173 in the new version. Thanks.

C: L 126. Unclear. Is 1 m the wavelength of radiation or dimension of the heater? R: This indicates the dimension of the heater (1.0 m long); and we have revised it accordingly in line 136.

C: L117-119: Suggest using concise and consistent treatment names. E.g. control, long term moderate warming, short-term acute warming. R: It has been done in line 129, and throughout the entire text.

C: L 283: "Mode" typo. R: Sorry for this mistake, it should be "model", and was corrected.

C: L 283: Please provide equation number. R: the equation number is 4, and was added.

C: L 238: Suggest different word for features R: We have changed to "belowground

characteristics" in line 248.

C: L 241-243: Suggest reporting an error estimate instead of range. R: This has been done in lines 252.

C: L 246: Define v/v R: it is defined as ratios of water volume and soil volume (added in the new revision, lines 255-256)

C: Throughout: Be consistent with significant figures (L264 : $R^2$ = 0.31 vs. L284: $R^2$ = 0.404 R: Thank you, we have revised R2 values with two 2 digits throughout the text.

Many thanks for the constructive comments and suggestions.

Please see the Manuscript-Revised-with supplement as Supplement (pdf).

Please also note the supplement to this comment:
https://www.biogeosciences-discuss.net/bg-2019-236/bg-2019-236-AC3-supplement.pdf

---

## Author Comment (AC5) · 2 Nov 2019

C: I forgot to add one minor textual edit to my review: Line 195: Replace were with was. The analysis (of variation) is singular and therefore requires a singular verb.

R: Thanks for the kind correction, it has changed to was already.

We greatly appreciate your constructive comments and kind suggestions which help us very much to improve our study, and hope our revisions and corrections would meet with your approval.

---

## Author Response (AR1)

Dear Dr. De Kauwe:

Thank you for your letter and the reviewers' comments concerning our manuscript entitled "Soil carbon release responses to long-term versus short-term climatic warming in an arid ecosystem". These valuable suggestions and comments help us greatly to improve our manuscript so that we have studied them carefully and made corrections point by point according to the constructive comments. The main changes in the revised manuscript have been highlighted using red font. Please see our point-by-point responses to your and the reviewers' comments in detail as following.

*C*: the original comments; *R*: the responses to the comments.

**Response to Dr. De Kauwe:**

*C*: I have read through the three reviews you've received as well as your manuscript and I have decided that major revisions are necessary. All of the reviewers were positive about your manuscript but they also suggested some important revisions.

*R*: Thank you for the positive comments.

*C*: I suspect there has been a bit of a misunderstanding about the Biogeosciences process. The process involves the authors responding with how they plan to revise the manuscript, awaiting the editor's decision and then submitting a revised manuscript alongside detailed point by point changes. In practice, you often have to revise the manuscript to do this and I can see that you've already attempted to upload a revised manuscript. As a result, I have looked through this and my sense is that the current changes are not yet sufficient.

*R*: Sorry for the misunderstanding about the Biogeosciences process. Yes, in the new revision, we have again revised and corrected the manuscript as kindly suggested. Please see details below.

***C***: The reviewers made some very specific suggestions about improvements about the clarity of experimental protocol and implications of results. I think more effort could be made in revising the text to accommodate these changes. In particular, I would like to see further discussion of the mechanisms and wider implications. I look forward to seeing these changes in the future.

***R***: More information on the experimental protocol was added (e.g., lines 130-137, 144-149, and line 244 as kindly suggested by the reviewers, red words). Particularly, more information on the mechanisms and implications were added in Discussion section. For the mechanisms, please see the lines 343-360, and 377-390. For the implications, please see lines 340-342, 364-366, 388-390; 421-425, and 465-467. This can strengthen our results. Many thanks for the kind suggestions.

***C***: In addition, one reviewer asked about data sharing. I refer the authors to the journals data availability statement https://www.biogeosciences.net/about/data_policy.html. I feel very strongly that all data should be shared in an open repository, I didn't find the existing statement sufficient. The journal specifically states: "If the data are not publicly accessible, a detailed explanation of why this is the case is required." I hope the authors will reconsider freely sharing their data.

***R***: Yes, we have shared the data in an open repository, the zenodo; and the statements has been added in the new version: "Data availability. The final derived data presented in this study are available at https://doi.org/10.5281/zenodo.3546062 (Yu et al., 2019)".

***C***: Finally, I see a comment directed at Dr. Bahn? This was confusing as they didn't review the manuscript, was this a mistake?

***R***: Sorry, this is a mistake. We should contact directly with you. Many thanks.

**Response to RC1:**

General Comments:

*C*: This paper specifically addresses our lack of knowledge on climate change within desert grassland ecosystems. While there have been many studies within boreal and temperate ecosystems, warming experiments focused on soil microbial community function and C stock assessments in arid systems are rare. Therefore, this study fills an important gap. The sampling design appears robust and is accompanied with a clear presentation of data. The methods for analyses are valid and informative enough for readers that might not be familiar with these techniques to understand, or where to read more, if desired. The authors further supported their hypotheses succinctly that warming reduced microbial respiration while wetting events enhanced respiration. However, my one concern is that 0.5 – 1.0 degree (Line 244) warming is not enough of a difference to define two treatments of warming (long-term, moderate and short-term, acute). Although, when moisture became limiting in August, there was a greater separation of the warming treatments which does highlight seemingly subtle differences. Perhaps a more direct addressing of the miniscule differences in the warming treatments until SWC becomes the greatest inhibitor to respiratory rates. Overall, I find this study well conducted and clearly presented. I would recommend for publication.

*R:* Thank for the positive comments. The larger differences occur between the two warming treatments and the control (i.e., ambient condition). Yes, acute warming treatments may not reach a higher level. However, it gradually induced a decline in soil moisture, finally limiting soil respiration. The major revision hade made carefully as kindly suggested (see detail below).

Specific Comments:

*C:* Line 12: Can examples of severe impacts be included here?

*R:* We have listed examples of severe impacts: the changes in litter decomposition and soil respiration in line 12-13 in the new revised version (red font). Thank you.

*C:* Lines 16-17: Since many readers do not get past an abstract, it would be useful to list treatment pressures in parantheses following long-term (ex.), short-term, etc.

*R:* Thanks. We have added the relevant information concerning the warming treatments in lines 16-17.

*C:* Line 19: Give the percentage of substantial water input treatment?

*R:* We fully irrigated the soil to field capacity and we have added it to this sentence lines 20-21. Many thanks.

*C:* Line 55: Use a more updated IPCC statistics (this one is 2014)

*R:* Thank you very much. But we are sorry about that because this information is from the last version (AR5 report of IPCC) that we can just find from IPCC. The future AR6 report has not been published (www.ipcc.ch/). However, we found a recent IPCC special report that highlights the warming-induced ecosystem degradation in relation to the study, and cited (IPCC 2019) (Line 59).

*C:* Warming treatment pressures were only applied during the growing season (June-Aug) of each year. In many temperate ecosystems, there has been evidence of seasonal extensions. Has there been any evidence that the growing season is becoming extended (earlier springs, later falls) at this site? If so, should this warrant extending warming treatments beyond June – August?

*R:* Thank you for your helpful advice. The growing season was not extended in our experimental site, an arid area, during our experiment. The warming-induced drought may limit the seasonal extensions, which may be outside our scope of the study. This may need to be explored in the future study in the arid ecosystem.

*C:* Line 774: If outlier point does not change equations, why not remove it completely?

*R:* Because it is the actual values that we measured; and the data have been included when the equations and their parameters were analyzed. Thus, it could be better that the points were shown. Additionally, we also presented the results excluded outlier points in Supplementary Figure S2.

Technical Comments:

*C:* Figure 2 line 769: Linear line is blue and not black as stated

*R:* Many thanks, we have corrected it accordingly.

*C:* Line 773: Typo? What does "soil animal" refer to?

*R:* Thanks for the kind comments. This is an improper phrase, and we have deleted it accordingly.

Thank you again for the valuable suggestions.

**Response to RC2:**

General Comments:

*C*: This study addresses an important research topical at climate change impacts on dryland soil carbon dynamics. This article presents valuable data from a field manipulation study in which the authors examined how warming and watering regimes of varying intensity and duration impact soil respiration in a desert steppe. While the study methods appear sound and the results provide strong evidence for warming-driven reductions in soil respiration, many sections in the text are unclear and need to be improved to strengthen and clarify the manuscript. The authors could modify hypothesis two into a statement that could be tested in this study and contribute to new insight on the dynamics of soil respiration in water-limited ecosystems.

*R*: Thank you for the positive comments. In the newest version, we have carefully revised and checked again the manuscript to strengthen and clarify the results as kindly suggested (please see the details below). The two hypotheses have been modified accordingly (Lines 106-110 of the newest version, highlighted by red words).

*C*: There are key findings that are not clearly reported and challenge my interpretation as a reader. Specifically, the authors should address an apparent conflict: warming decreased $R_s$ despite the positive relationship between $R_s$ and soil temperature. The authors should explicitly highlight the important role of soil moisture as the dominant control on $R_s$ rates and temperature sensitivity.

*R*: Thank you for helpful comments. Actually, that is, the persistent warming treatments decreased average $R_s$. The positive relationship between $R_s$ and soil temperature occurs in each plot or each treatment. The two data sets are different, the former is continued warming treatment effect (comparison among the treatments: long-term warming, short-term warming, and ambient as a control), and the latter is the relationship between $R_s$ and soil temperature. Yes, the important roles of soil moisture as the dominant control on $R_s$ rates and temperature sensitivity were highlighted in many appropriate places of the newly revised version (e.g., lines, 22-23, 100-101, 377-390).

*C*: Lastly, the data availability statement does not appear to meet the journal's data policy requirements, and I suggest uploading data to a public repository, if possible.

*R*: Many thanks. In the newest version, we have shared the data in an open repository, the zenodo; and the statements has been added in the new version: "Data availability. The final derived data presented in this study are available at https://doi.org/10.5281/zenodo.3546062 (Yu et al., 2019)".

Specific comments: Parts of this manuscript would benefit from additional explanation. Below I provide some specific examples.

*C*: L 24-27. "This indicates that soil carbon release responses strongly depend on the duration and magnitude of climatic warming, which may be driven by SWC and soil temperature." This is unclear. Please explain how SWC and soil temperature influence soil respiration, and then perhaps infer how those relationships have implications for climatic warming impacts on soil carbon dynamics.

*R*: Thank you, we have revised it to "This indicates that climatic warming constrains soil carbon release, which is controlled mainly by decreased soil moisture, consequently influencing soil carbon dynamics" to be clearer and more concise (Lines 21-23 of the newly revised version). The relevant mechanism has been added as kindly suggested (Lines 343-360).

*C*: L 55-59: An explanation of why low precipitation and biomass enhances vulnerability would strengthen the authors' claim that deserts are sensitive to climate change.

*R*: Many thanks. We have made it to be clearer and concise accordingly, and the explanation was also added accordingly: "For instance, water deficit and heat waves during growing season can markedly decrease plant cover and productivity in this arid ecosystem" (Lines 59-63).

*C*: L 60-66: This section shows that temperature and moisture are well-known controls on $R_s$. However, this conflicts with the previous claim (L43-47) that Rs responses to biotic and abiotic factors are poorly understood. Can this apparent contradiction be addressed in a way that makes a stronger case for this study? E.g. whereas soil moisture and temperature are well-known controls on $R_s$, it is not well known how soil moisture modulates the response of $R_s$ to changes in the duration and intensity of warming.

*R*: we have changed the expressions in both sections to be clearer and more logical (Lines 44-48; 66-68). Many thanks.

*C*: L 84: Please elaborate on "undefined" since many studies have reported $R_s$ pulses after water inputs (Huxman et al., 2004; Sponseller, 2007). Huxman, Travis E., et al. "Precipitation pulses and carbon fluxes in semiarid and arid ecosystems." Oecologia 141.2 (2004): 254-268. Sponseller, Ryan A. "Precipitation pulses and soil $CO_2$ flux in a Sonoran Desert ecosystem." Global Change Biology 13.2 (2007): 426-436.

*R*: Thank you for the useful advice, this part has been revised accordingly: the inappropriate word "undefined" has been removed. And we cited the two references (Lines 90-93).

*C*: L 86-88. This argument would be stronger if the authors explained why a long-term study (4 years) might yield insights undetected in previous two-year studies. Why do the authors expect to find something new?

*R*: Thank you. Yes, we have important findings in previous two-year which have been published (Liu et al. Plant Soil, 2016, 400:15–27). In the current study, however, we expect that the long-term (four-year) warming have different effects on $R_s$ (i.e., more profound, even reverse effects relative to previous two-year short term); and the underlying mechanism under longer term warming condition, and the role of soil water status to $R_s$ responses to climatic warming, are also uncertain (added this explanation in the new version, lines 97-101).

*C*: L 88-89: Unclear. Please elaborate.

*R*: Thanks very much, we have re-edited it to "and the underlying mechanism under longer term warming condition, and the role of soil water status to $R_s$ responses to climatic warming, are also uncertain" (also see above).

*C*: L 97-98: The introduction section already provides evidence in support of H2. In its current form, it is not clear why it is worth testing H2 in this study. How could H2 be modified into a hypothesis that could be tested in this study and contribute to new insight on the dynamics of soil respiration in water-limited ecosystems?

*R*: This H2 has been modified to "the dynamics of $R_s$ in the water-limited ecosystem can be driven mainly by the combination of soil temperature and soil moisture, and soil moisture can modulate the response of $R_s$ to warming". Many thanks.

*C*: Results 3.1. Warming effects on soil features L 251-254: According to the Supplementary Table S1, belowground biomass is 11.5 units for the Acutely Warmed treatment. Is this a typo? It is considerably higher than the BB reported for other treatments.

*R*: Thanks for your comments. This is a mistake, it should be 1.15, and we have corrected it (Supplementary Table S1).

*C*: 3.2: It is unclear why this is section is titled "Watering pulse effects on $R_s$." Does this section refer to data collected only after watering? Or does the section report findings from all measurement dates?

*R*: We have two experiments: one is the warming experiment which included three treatments: control, long-term moderate warming, and short-term acute warming. The other is the watering pulse treatments which included control and watering treatment to further highlight the important role of water status. Yes, this section referred to data collected in the plots of watering treatments.

*C*: Figure 2: Please explain the data source – do the data represent the control or warmed treatments? Also, is it necessary to show the linear and quadratic fits? Are these pieces of information reported or used to make inferences?

*R*: This section mainly focused on the relationship between $R_s$ and soil water content. The data were collected in the plots of watering treatments (added in the figure 2 legend of the new version), and were used to determine the relationship between $R_s$ and soil water content in dessert steppe. Here, we focused on the comparisons between the linear, quadratic, and Gompertz functional models. Thus, the information used could be useful. Many thanks for the kind comments.

C: Figure 3A. This figure presents information that is critical for the authors' conclusion. It provides evidence for why $R_s$ was lower in warmed treatments, despite having a positive relationship with soil temperature. I suggest leading Section 3.2 or 3.3 with a strong statement describing the relationship between $R_s$, temperature, and moisture. For example, soil respiration increased exponentially with temperature in watered plots but was lower and insensitive to temperature in the control plots.

*R*: Thanks for your useful advice, it has been revised accordingly in lines 294-298.

*C*: L 771: Unclear. What is the initial $R_s$ response to SWC? What do the other points represent?

*R*: It should be linear $R_s$ response to SWC at low levels. This is Gompertz functional model features: for the all points, with SWC increasing, $R_s$ linearly increased sharply, then reaching a maximum value, and levelling off at a stable level. The relevant explanations have been added accordingly (Lines 377-390). Many thanks.

*C*: Section 3.3 Suggest leading with conclusive evidence. For example, "Warming regimes resulted in marked declines in $R_s$. Whereas no difference in Rs was observed in July, during August average $R_s$ values were x, y, z for the control, moderately warmed, and acutely warmed treatments, respectively."

*R*: Thanks, it has been done in lines 285-288.

*C*: Section 3.4 needs a figure reference.

*R*: The reference figure is figure 5, and was added (Line 303).

*C*: This section should explain why $R_s$ decreased in warmed plots despite having a positive relationship with soil temperature.

*R*: They are different two terms: $R_s$ in warmed plots were the values averaged in the warming treatments, whereas $R_s$ values used for the relationship with soil temperature are the data in each plots or each treatments; and particularly, the soil temperature data used for the relationship $R_s$ and soil moisture are the values when the $R_s$ were measured simultaneously. They two are matching values each other. Thus, long-term warming rather than temporary high temperature reduced $R_s$, despite having a positive relationship with soil temperature (also added, lines 294-298).

*C*: L 319-322: Unclear how $R_s$ can acclimate to warming but also decrease. Please explain the mechanism. Is the acclimation referring to changes in microbial respiration? Are net reductions in $R_s$ driven by temperature-stress impacts on plant and root activity?

*R*: $R_s$ can acclimate to warming but also decrease. The acclimation refers to changes in both root and microbial respiration. A net reduction in $R_s$ may be partly driven by temperature-stress impacts on root activity (e.g., the continual warming can limit the root activity, thus reducing $R_s$). The "plant" has been removed because this study mainly focused on the belowground parts. This section has been substantially revised, the relevant mechanism has been added as kindly suggested (Lines 343-360, red parts): "Actually, the $R_s$ [the sum of root (autotrophic, $R_a$) and

$R_h$ respiration–the former accounting for *c.* 22 % of the total $R_s$ in the ecosystem, Liu et al. 2016] may acclimatize to warming within an appropriate range of temperature change at an ample soil moisture; however, it decreases with increasing temperatures above an optimum level. The mechanisms may include: within an appropriate range of temperature change at an ample soil moisture, climatic warming can enhance both plant root (Luo et al., 2001; Liu et al. 2016) and microbial activities (Tuker et al. 2014), leading to increases in both $R_a$ and $R_h$ , consequently the $R_s$ (Luo et al., 2001; Tuker et al. 2014; Xu et al., 2019). However, when warming continues or with increasing temperatures above an optimum level, the root growth can be constrained, directly reducing $R_a$ (Carey et al., 2017; Liu et al., 2016; Luo et al., 2001; Wan et al., 2007); and the limitation to microbial activities may also occur (Tucker et al.. 2013; Yu et al., 2018), decreasing the $R_h$ (Bérard et al., 2011; Tucker et al., 2013; Bérard et al., 2015; Romero-Olivares et al., 2017). In addition, decreases in soil enzyme pools and its activity under warming may also contribute to a reduction in $R_a$ (e.g., Alvarez et al. 2018). Further, $R_s$ decreases with warming under water deficit (Moyano et al., 2013; Wang et al., 2014; and see below). Together, the declines in both root and microbial respirations finally reduce the $R_s$". Many thanks for the valuable comments.

*C*: L358-362: Consider citing previous studies documenting that the temperature response of $R_s$ is conditional on moisture (Roby et al., 2019; Conant et al., 2000). Roby, M. C., Scott, R. L., Barron-Gafford, G. A., Hamerlynck, E. P., & Moore, D. J. (2019). Environmental and Vegetative Controls on Soil CO2 Efflux in Three Semiarid Ecosystems. Soil Systems, 3(1), 6. Conant, Richard T., Jeffrey M. Klopatek, and Carole C. Klopatek. "Environmental factors controlling soil respiration in three semiarid ecosystems." Soil Science Society of America Journal 64.1 (2000): 383-390.

*R*: Thank you, we have cited them already in lines 416-417, and added in the reference list (red parts).

TECHNICAL COMMENTS

**C:** L22: Features is unclear.

**R:** This has been changed it to "The belowground biomass, soil nutrition, and microbial biomass" to detail these soil variables in line 26 of the new revision.

**C:** L 143: What are the units of soil moisture?

**R:** It has been added (a ratio: v/v) (Line 160).

**C:** L 227: Please provide depth of soil temperature measurements.

**R:** We have provided it in line 244.

**C:** L 199: First mention of SWC; please define or introduce this acronym in section 2.3

**R:** the SWC whole name has been added in line 180 in the new version. Thanks.

**C:** L 126. Unclear. Is 1 m the wavelength of radiation or dimension of the heater?

**R:** This indicates the dimension of the heater (1.0 m long); and we have revised it accordingly in line 141.

**C:** L117-119: Suggest using concise and consistent treatment names. E.g. control, long term moderate warming, short-term acute warming.

**R:** It has been done in line 130-131, and throughout the entire text.

**C:** L 283: "Mode" typo.

**R:** Sorry for this mistake, it should be "model", and was corrected.

***C***: L 283: Please provide equation number.

***R***: the equation number is 4, and was added (Line 303).

***C***: L 238: Suggest different word for features

***R***: We have changed to "belowground characteristics" in line 255.

***C***: L 241-243: Suggest reporting an error estimate instead of range.

***R***: This has been done in lines 259.

***C***: L 246: Define v/v

***R***: it is defined as ratios of water volume and soil volume (added in the new revision, lines 262-263)

***C***: Throughout: Be consistent with significant figures (L264 : $R^2 = 0.31$ vs. L284: $R^2 = 0.404$

***R***: Thank you, we have revised $R^2$ values with two 2 digits throughout the text. Many thanks for the constructive comments and suggestions.

**Response to RC3:**

General Comments:

***C***: The paper describes a four year warming and wetting experiment in a desert steppe in Northern China. The introduction gives a good overview of the latest and more established scientific insights and the authors did a thorough measurement campaign. I particularly appreciate how much work went into the various additional belowground measurements. Given the limited number of such experiments for this ecosystem type, this work is certainly of interest to readers of Biogeosciences. Overall, the paper is well structured and written clearly, but I feel there are some elements in the text that require clarification or some more in-depth information. If the authors manage to improve these elements I would recommend the paper for publication.

*R*: Thank for the positive comments. The manuscript has been revised as kindly suggested.

**Major comments:**

*C***:** My first major comment is directly about the abstract. There is a seemingly counterintuitive message there that confused me when reading it: Long-term warming reduced $R_s$ by 32.5 percent (line 18). Yet, long term climatic warming decreased SOC (line 24)? While this is certainly possible, it is not directly what one would expect. Was this reduction in SOC caused by an initial spike in $R_s$ at the beginning of the experiment? The lower SOC content could then also contribute to decreasing respiration due to reduced availability of substrate to microbes to decompose. Yet, the authors mainly talk about the moisture effect and how low soil moisture decreased $R_s$. The mentioned decline in SOC from the abstract is not presented in the results and discussion. Actually, the authors state that "in the present study, SOC concentrations were not significantly affected by climatic warming" and then later write "although SOC might be expected to decrease with long-term climatic warming" (conclusion iv). I do not understand how such a strong statement can be made in the abstract when the results and discussion show otherwise and even contradict one another. Given the high number of people that generally do not read beyond the abstract, my suggestion is to 1) rewrite this part of the abstract more clearly and 2) to present the evidence to support this claim more clearly in the results and discussion.

*R*: This suggestion is very valuable for us. To be clearer and more logical throughout the text, we have rewritten the relevant expressions: e.g., we deleted the "soil organic carbon content tended to decrease with long-term climatic warming" because of the results: "SOC concentrations were not significantly affected by climatic warming (Supplementary Table S1). Yes, based on the present results, we mainly focused on the new significant findings' aspects: the long-term warming effects on soil preparation, watering effects, and the its relationships with soil moisture and soil temperature. Thank you for the kind suggestions.

*C*: My second major comment is about the authors' choice for the various model fittings in the statistical analysis and in particular for the Gompertz function. The authors provide limited explanation for choosing the Gompertz equation in section 4.2, line 334-337 and mention the parabolic curve function as another viable option. Indeed, in section 3.2 there is another model with a better fit: the quadratic functional model. The authors do not argue further why they still continued parameter fitting with the Gompertz curve, despite the quadratic model having a seemingly better fit (figure 2 and section 3.2). I would like to know 1) why the Gompertz function was selected and 2) how picking that curve to fit the parameters for the non-linear regression model (eq 4) affected the results compared to taking the parameters from a quadratic model fitting (sensitivity analysis)?

*R*: We conducted the Gompertz relationship to clarify the relationship because the data most likely support a Gompertz (i.e., saturating, sigmoidal) relationship rather than a linear relationship. The parabolic curve mentioned (in section 4.2) is inappropriate, was deleted. 1) the Gompertz relationship can well fit with the relationships between $R_s$ and soil water content ($R^2 = 0.87$; RMSE = 4.88; also refer to e.g., Gompertz, 1825; Yin et al., 2003), which also can obtain some key thresholds (e.g., the asymptote value, the optimal SWC) that can not obtain from both linear and quadratic functional models. 2) A non-linear regression model (eq 4) is used to fit the relationship of $R_s$ with both soil temperature and soil moisture. The optimal SWC of 0.229 (v/v) was estimated by the Gompertz functional curve. This optimal SWC (means that a SWC value when $R_s$ reach a maximum) is a necessary parameter of equation 4, which is just obtained the Gompertz functional model. Thanks for the valuable comments.

**Minor comments:**

*C*: The Gompertz function (line 22): This function (and its shape) might not be a given knowledge for all readers. My suggestion is to rephrase in the abstract to "whereas the relationship between $R_s$ and soil moisture was better fitted to a sigmoid function" and explain the Gompertz curve further in Section 2.6 (see major comment #2).

*R*: Thank you, this has been done in line 25 the abstract and line 229 in Section 2.6. From the sigmoid function curve, the thresholds of the changes in Rs with increasing SWC can be obtained (236-237).

*C*: Line 48: The desert steppe is c. 8.8 million square hm. Do the authors mean total global desert steppe area or the area in China?

*R*: It means the area in China, and has been revised to "The desert steppe of China" (Line 51). It should be *c.* 8.8 million square hm, and was corrected. Many thanks.

*C*: Line 74/75: I would suggest adding the more recent reference to Yan et al. 2018 here as well.

*R*: Thanks, we have added it already in line 82, and in the reference list.

*C*: Reference:

Yan, Z., B. Bond-Lamberty, K. E. Todd-Brown, V. L. Bailey, S. Li, C. Liu, and C. Liu (2018), A moisture function of soil heterotrophic respiration that incorporates microscale processes, Nature Communications, 9(1), 2562, doi: 10.1038/s41467-018-04971-6.

*R*: It has been cited and added in the reference list.

Many thanks for the constructive comments and suggestions.

**Anonymous Referee #4:**

*C*: I forgot to add one minor textual edit to my review:

Line 195: Replace were with was. The analysis (of variation) is singular and therefore requires a singular verb.

*R*: Thanks for the kind correction, it has been done.

We greatly appreciate your constructive comments and kind suggestions which help us very much to improve our study, and hope our revisions and corrections would meet with your approval.

[revised manuscript text omitted]

---

## Referee Report (RR1)

**Reviewer's Final Report**

**Author(s):** Yu et al.

**Title:** Soil carbon release responses to long-term versus short-term climatic warming in an arid ecosystem

**General Comments**:

The authors worked hard to address the concerns in the first round of critiques, and it shows here. My only commentary is for clarifying the language of the authors' sections of new text. Please see below for specific examples and suggestions for edits.

**Specific Technical Comments**:

Line 12-13: Edit out "such as the changes in" and replace with 'by altering rates of' for a stronger opening sentence

Line 16: Remove "about" for ~3°C

Line 17: Remove "about" for ~4°C

Line 30: precipitation pulse**s** during the growing season

Line 37: of the earth's

Line 62: during the growing season

Lines 94-101: A previously published warming experiment is referenced but no explicit mention of those results are given. The new, long-term warming experiment is then stated to possibly have an opposite result, but again, no explicit statement of what that might be. Could the authors clarify this paragraph?

Line 104: Remove "about" for ~3 and ~4°C, or since this is clarified in the Abstract and then outlined in the Methods a few paragraphs later, this detail could be omitted here.

Line 132: remove long-term as this sentence (line 131 above) started by defining this as the longer warmed plots

Line 137: space needed between regimes (one-year)

Line 148: Can remove the sentence defining the control plots as they were already defined in lines 138-139

Line 150: were installed with a "dummy" heater…

Line 274: Respiration significantly increased with…

Line 290: end sentence with: from each other (P= 0.45)

Line 340-342: Awkwardly worded sentence. Rework to state that the positive feedback loop could be weakened with length or intensity of warming

Line 343: Edit out Actually, and start sentence with Total respiration (Rs)

Line 350: add in "and" before consequently the total Rs

Line 378: may rapidly increase

Line 382-384: Edit out "at a higher level" as substrate limitation is just another limitation, not necessarily higher (or more important) than moisture, or soil type for instance. Unless you mean higher level as in greater respiration….Please clarify

Line 385: high meaning increased respiration?

Line 414: thus affecting its temperature sensitivity; SWC becomes the main factor…

Line 419: warming; and – split into 2 full sentences

Line 421: implicate that multiple factors together…

---

## Author Response (AR2)

Dear Dr. De Kauwe:

Thank you for the kind consideration. We have carefully checked and revised the manuscript entitled "Soil carbon release responses to long-term versus short-term climatic warming in an arid ecosystem" (bg-2019-236) based on your and the reviewers' suggestions and comments.

We also checked and corrected it carefully for other minor issues such as typos, format style of the journal. The main changes in the newly revised manuscript have been highlighted using red font (the clean version) or the track changes mode (a marked-up manuscript version at the end of this letter). Please see our point-by-point responses as following.

*C*: the original comments; *R*: the responses to the comments.

**Response to Dr. De Kauwe:**

*C*: Two reviewers have now seen your revised manuscript. I'm happy to recommend acceptance subject to a few minor suggested revisions from the reviewers.

*R*: Thank you for the kind consideration. The manuscript has been carefully revised and corrected accordingly.

**Response to R1:**

*C*: My only commentary is for clarifying the language of the authors' sections of new text.

Please see below for specific examples and suggestions for edits.

*R*: Thank you for the kind comments. Yes, we have revised the relevant sections as kindly suggested, and carefully checked and corrected the entire text.

**Specific Technical Comments:**

*C*: Line 12-13: Edit out "such as the changes in" and replace with 'by altering rates of' for a stronger opening sentence

*R*: This has been done (please see the red words in the new version; lines 12-13).

*C*: Line 16: Remove "about" for ~3°C

**R:** It has been done. Many thanks.

**C:** Line 17: Remove "about" for ~4°C

**R:** It has been done.

**C:** Line 30: precipitation pulses during the growing season

**R:** It has been done.

**C:** Line 37: of the earth's

**R:** It has been done.

**C:** Line 62: during the growing season

**R:** It has been done.

**C:** Lines 94-101: A previously published warming experiment is referenced but no explicit mention of those results are given. The new, long-term warming experiment is then stated to possibly have an opposite result, but again, no explicit statement of what that might be. Could the authors clarify this paragraph?

**R:** We added the relevant information to be clarified: "A previous study has indicated that the short-term (two-year) warming (2°C) did not affect significantly respiration rate during the growing season" and revised this paragraph accordingly (Lines 94-101 in the newest version).

**C:** Line 104: Remove "about" for ~3 and ~4°C, or since this is clarified in the Abstract and then outlined in the Methods a few paragraphs later, this detail could be omitted here.

**R:** Yes, we agree to delete the information here since this is clarified in the Abstract and then outlined in the Methods later. Many thanks.

**C:** Line 132: remove long-term as this sentence (line 131 above) started by defining this as the longer warmed plots

**R:** Yes, we agree to revise this sentence, and it has been done.

**C:** Line 137: space needed between regimes (one-year)

**R:** It has been done.

**C:** Line 148: Can remove the sentence defining the control plots as they were already defined in lines 138-139

**R:** It has been removed. Thank you.

Line 150: were installed with a "dummy" heater...

**R:** It has been done.

**C:** Line 274: Respiration significantly increased with...

**R:** It has been done.

**C:** Line 290: end sentence with: from each other (P= 0.45)

**R:** The phrase has been added.

**C:** Line 340-342: Awkwardly worded sentence. Rework to state that the positive feedback loop could be weakened with length or intensity of warming

**R:** The phrase has been revised as kind suggested (Lines 340-341 in the new version).

**C:** Line 343: Edit out Actually, and start sentence with Total respiration (Rs)

**R:** This has been done.

**C:** Line 350: add in "and" before consequently the total Rs

**R:** Yes, the "and" has been added.

**C:** Line 378: may rapidly increase

**R:** This has been done.

**C:** Line 382-384: Edit out "at a higher level" as substrate limitation is just another limitation, not necessarily higher (or more important) than moisture, or soil type for instance. Unless you mean higher level as in greater respiration....Please clarify

**R:** We agree to edit out the phrase "at a higher level" as kindly suggested.

**C:** Line 385: high meaning increased respiration?

**R:** Yes, the "high" means "increased", and it was changed accordingly.

**C:** Line 414: thus affecting its temperature sensitivity; SWC becomes the main factor...

**R:** This has been done.

**C:** Line 419: warming; and – split into 2 full sentences

**R:** This has been done.

**C:** Line 421: implicate that multiple factors together...

**R:** This has been done.

Many thanks for the constructive suggestions and comments.

**Response to R2:**

**C:** Consider including the hypotheses in the abstract.

**R:** The hypotheses could not be necessary in the abstract. However, we revised the expression to be more specific and stronger as kind suggested (bellow).

**C:** Hypothesis 1 can be more specific. Currently, the hypothesis states that "soil moisture plants a key factor controlling Rs." This hypothesis would be stronger if the author's specified how soil moisture is expected to control Rs.

**R:** Yes, for the hypotheses 1, we changed to "decreased soil moisture plays a key role in reducing $R_s$". This can be more specific and stronger (Lines 107-108).

**C:** I do not necessarily feel you need to include your hypotheses in the abstract that is up to you, but I do think you could revise the hypotheses so that they are more insightful, I think the reviewer makes a good point.

**R:** Yes, the hypotheses could not be necessary in the abstract. And we have revised this expression for the hypotheses as kindly suggested.

Many thanks for the constructive suggestions and comments.

Please see a marked-up manuscript version below:

[revised manuscript text omitted]